# Estimating Criteria Pollutant Emissions Using the California Regional Multisector Air Quality Emissions (CA-REMARQUE) Model v1.0

Christina B. Zapata[1], Chris Yang[2], Sonia Yeh[2], Joan Ogden[2], Michael J. Kleeman[1]

[1] Department of Civil and Environmental Engineering, University of California – Davis, Davis, California, USA
[2] Institute of Transportation Studies, University of California – Davis, Davis, California, USA

Correspondence to: Michael J. Kleeman (mjkleeman@ucdavis.edu)

**Abstract.** The California REgional Multisector AiR QUality Emissions (CA-REMARQUE) model is developed to predict changes to criteria pollutant emissions inventories in California in response to sophisticated programs implemented to achieve deep Green House Gas (GHG) emissions reductions. Two scenarios for the year 2050 act as the starting point for calculations: a Business as Usual (BAU) scenario and an aggressive GHG reduction (GHG-Step) scenario. Each of these scenarios was developed with an energy economic model to optimize costs across the entire California economy and so they necessarily include changes in activity, fuels, and technology. Separate algorithms are developed to estimate emissions of criteria pollutants (or their precursors) that are consistent with the future GHG scenarios for the following economic sectors: (i) on-road, (ii) rail and off-road, (iii) marine and aviation, (iv) residential and commercial, (v) electricity generation, and (vi) biorefineries. Properly accounting for new technologies involving electrification, bio-fuels, and hydrogen plays a central role in these calculations. Critically, criteria pollutant emissions do not decrease uniformly across all sectors of the economy. Emissions of certain criteria pollutants (or their precursors) increase in some sectors as part of the overall optimization within each of the scenarios. This produces non-uniform changes to criteria pollutant emissions in close proximity to heavily populated regions when viewed at 4km spatial resolution, with obvious implications for exposure to air pollution for those populations. As a further complication, changing fuels and technology also modify the composition of reactive organic gas emissions and the size and composition of particulate matter emissions. This manifests most notably through a comparison of emissions reductions for different size fractions of primary particulate matter. Primary PM2.5 emissions decrease by 4% in the GHG-Step scenario vs. the BAU scenario while corresponding primary PM0.1 emissions decrease by a factor of 36%. Ultrafine particles (PM0.1) are an emerging pollutant of concern expected to impact public health in future scenarios. The complexity of this situation illustrates the need for realistic treatment of criteria pollutant emissions inventories linked to GHG emissions policies designed for fully developed countries and states with strict existing environmental regulations.

## 1 Introduction

The United States, along with many developing countries, is debating optimal strategies to mitigate threats to long-term prosperity including (among other things) climate change and threats to public health. These specific issues are

at least partially linked through regional air quality. Realistic mitigation plans for Green House Gas (GHG) emissions ($CO_2$, $CH_4$, $N_2O$, etc) usually include measures encouraging reduced energy consumption or changes to energy sources leading to reduced GHG emissions. These measures also impact emissions of criteria pollutants or their precursors (PM, NOx, SOx, VOCs, $NH_3$, etc) that influence regional air quality. Air quality influences public health through impacts on mortality (primarily related to $PM_{2.5}$) and morbidity (primarily related to $PM_{2.5}$ and $O_3$).

Many previous attempts to characterize the impact of climate policies on criteria pollutant emissions, air quality, and public health have often emphasized countries where potential health savings are largest. These previous studies have also usually performed calculations for large geographic areas without resolving details at regional scales appropriate for California (Bollen, van der Zwaan et al. 2009, van Aardenne, Dentener et al. 2010, Rafaj, Schöpp et al. 2012, Shindell, Kuylenstierna et al. 2012, West, Smith et al. 2013, Garcia-Menendez, Saari et al. 2015). These studies represent California with only a small number of grid cells or they uses simplistic representations of California's energy economy.

More recent studies addressing interactions between climate policies, emissions, and air quality in the US (Loughlin, Benjey et al. 2011, Ran, Loughlin et al. 2015, Rudokas, Miller et al. 2015, Trail, Tsimpidi et al. 2015, Zhang, Bowden et al. 2016, Keshavarzmohammadian, Henze et al. 2017) have allocated future emissions using enhanced population surrogates (Ran, Loughlin et al. 2015) and federal climate policies (Trail, Tsimpidi et al. 2015). The current study builds on this previous work to explicitly account for California's ambitious climate regulations broken out by detailed sectors including realistic siting of biofuel facilities. The current study also considers the effects of regenerative braking, and exhaust particulate size and speciation changes from the heavy use of alternative and renewable fuels across multiple economic sectors. These enhancements support the desired level of detailed analysis for the intersection of air, climate, and energy choices in California.

The purpose of this paper is to describe the California REgional Multisector AiR QUality Emissions (CA-REMARQUE) model that can translate complex GHG mitigation scenarios to criteria pollutant emissions inventories with sufficient detail to support fine-scale air quality models and public health analysis. Here we emphasizes solutions that optimize state-wide total GHG emissions across the entire California economy, with potential tradeoffs between different source types to achieve this objective. The complex optimization problem requires an energy economic model, and so we focus on scenarios predicted by the CA-TIMES energy economic model as the starting point for the analysis. The detailed algorithms within the CA-REMARQUE model are then developed to translate predicted changes in GHG emissions associated with source activity, fuels, and technology to criteria pollutant emissions that are spatially-resolved (4 km) for each sector of the California economy. Changing emissions profiles caused by fuel substitutions are also accounted for. Final results are compared to an expert-analysis method developed for a previous global analysis to illustrate why the complex methods described in this study are needed when analysing developed regions like California that have major diversified economies and a long history of environmental regulations.

## 2 Methodology

Energy scenarios are translated to criteria pollutant emissions inventories by the CA-REMARQUE model in a multi-step process with unique algorithms developed for each major sector of the economy that emits air pollution precursors. All calculations start with energy scenarios developed by the energy economic model CA-TIMES. The details needed to produce criteria pollutant emissions inventories are discussed in the following sections.

### 2.1 CA-TIMES Energy Model and Energy Scenarios

CA-TIMES (McCollum, Yang et al. 2012, Yang, Yeh et al. 2014, Yang, Yeh et al. 2015) is a bottom-up energy-economic model originally based on the MARKAL TIMES model (Loulou, Goldstein et al. 2016). CA-TIMES is a cost-minimization optimization model that balances energy supply and demand system-wide from all economic sectors of the energy economy. Demand sectors include transportation, industrial, residential, commercial, and agricultural. Fuel and electricity supply includes electric, biofuel, hydrogen production plants and biofuel and petroleum refineries. Demand was assumed fixed for the scenarios considered (Yang, Yeh et al. 2014, Yang, Yeh et al. 2015). CA-TIMES allows imports from out of state, such as oil, natural gas, and electricity. Renewables and Biomass are handled separately and modelled explicitly as located in or out of state and imports are determined on a cost basis. CA-TIMES contains capital and operation costs for each technology, diverse fuel and energy carriers, and calculates GHG emissions for $CO_2$, $CH_4$, and $N_2O$.

The case studies considered in the present study focus on two CA-TIMES scenarios in 2050: (i) a Business as Usual (BAU) scenario that achieves the goals outlined in California Assembly Bill 32 (AB32), the Global Warming Solutions Act of 2006 and (ii) a climate friendly GHG-Step scenario that achieves an 80% reduction (relative to 1990 levels) in GHG emissions by 2050. In the GHG-Step scenario a "step" GHG emissions constraint is applied in which a constant 2020 cap is held until 2050, and then an 80% reduction is applied from 2050 onward. This allows the model freedom to adopt strategies that lower GHG emissions prior to 2049 if those strategies minimize costs. This 2050 GHG constraint does not shock to the energy system because the CA-TIMES model has perfect foresight and optimally minimizes the energy system cost (with a 4% discount factor) over the entire period from 2010 to 2050 making investment decisions to meet targets. Also, CA-TIMES investments in low-GHG technologies start slowly and grow to reach the required market share to meet the targets since technologies have finite lifetimes and cannot take over respective markets instantaneously. The criteria pollutant emissions between 2010 and 2049 were not analysed in the current study but a summary of CA-TIMES results for intermediate years is provided by (Yang, Yeh et al. 2015). Both BAU and GHG-Step scenarios include current and sunset GHG regulations in California (Corporate Average Fuel Economy (CAFE) Standards (California Air Resources Board 2005, California Air Resources Board 2009, California Air Resources Board 2010), Zero Emission Vehicle (ZEV) Mandate (California Air Resources Board 2012, California Air Resources Board 2012, California Air Resources Board 2012, California Air Resources Board 2012, California Air Resources Board 2012), Low Carbon Fuel Standard (LCFS) (California Air Resources Board 2009, California Air Resources Board 2011), Cap-and-Trade Program (California Air Resources Board 2011, California Air Resources Board 2017) and federal and state incentives (tax credits and subsidies). CA-TIMES predicts total annual

energy consumption in California for the year 2050 to be 8,763 PJ in the BAU scenario and 7,679 PJ in the GHG-Step
scenario (reference value for 2010 is approximately 7,500 PJ) (Yang, Yeh et al. 2015).
The methods to estimate criteria emissions for different sources developed in the current paper take advantage of the
best available information describing future energy and emissions as a function of location.  The quality of this
information varied considerably for each major source category and so the details of the methodology also varied.
Figure 1 illustrates an overview of the general procedure.  The changes in energy consumption and GHG emissions
produced by CA-TIMES for each energy sector in the year 2050 were translated to changes in criteria pollutant
emissions by accounting for changing energy activity levels or fuel switching.  Literature searches were conducted to
identify any previous studies describing spatial locations of future emissions within California.  Altered emissions for
the year 2050 were then projected from a 2010 emissions inventory with 4 km spatial resolution provided by the
California Air Resources Board (CARB).  Additional details for each major source type are discussed below.

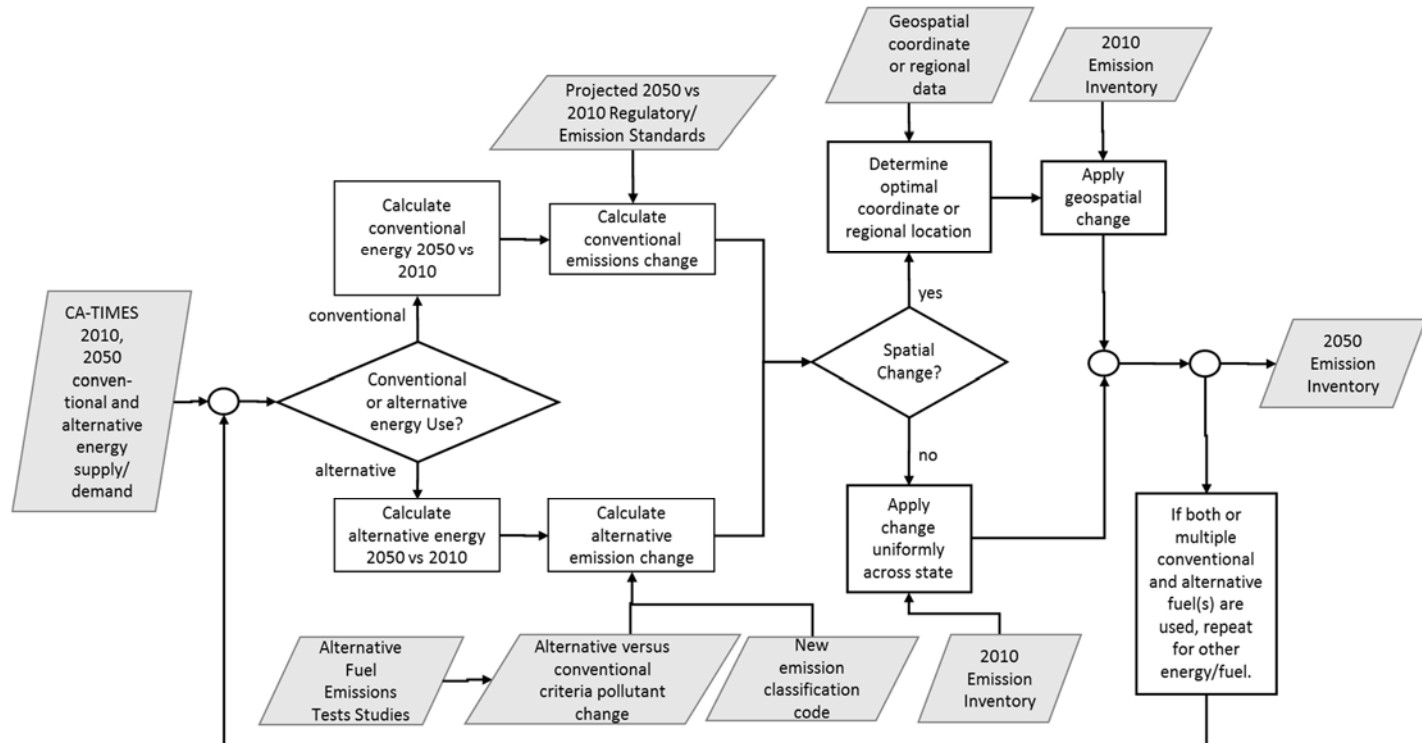


**Figure 1: Process diagram of emission inventory generation for each sector or mode.**

**2.2 CA-REMARQUE On-road Mobile Algorithms**
On-road mobile sources include passenger cars, light duty trucks (LDT), medium duty trucks (MDT), heavy duty
trucks (HDT), buses, motorcycles, and motor homes.  On-road emissions were generated in a multi-step process
summarized in Fig. 2.  In the first step, 2010-2035 emission projection trends from the EMission FACtor (EMFAC)
2011 model (California Air Resources Board 2011) were used to extrapolate further to 2050.  In the second step, an
intermediate 4km vehicular emissions inventory was generated by combining EMFAC 2050 projections with 2010
4km emission inventory as a spatial surrogate.  In the third step, the 2050 fossil fuel vehicular emission rates that were
projected from EMFAC as well as new emission rates gathered from alternative fuel emission literature were used to
scale the 4km intermediate mobile emission inventory based on the vehicle miles travelled (VMT), trips, and vehicle
class and (conventional and alternative) fuel consumption output produced for each CA-TIMES scenario.

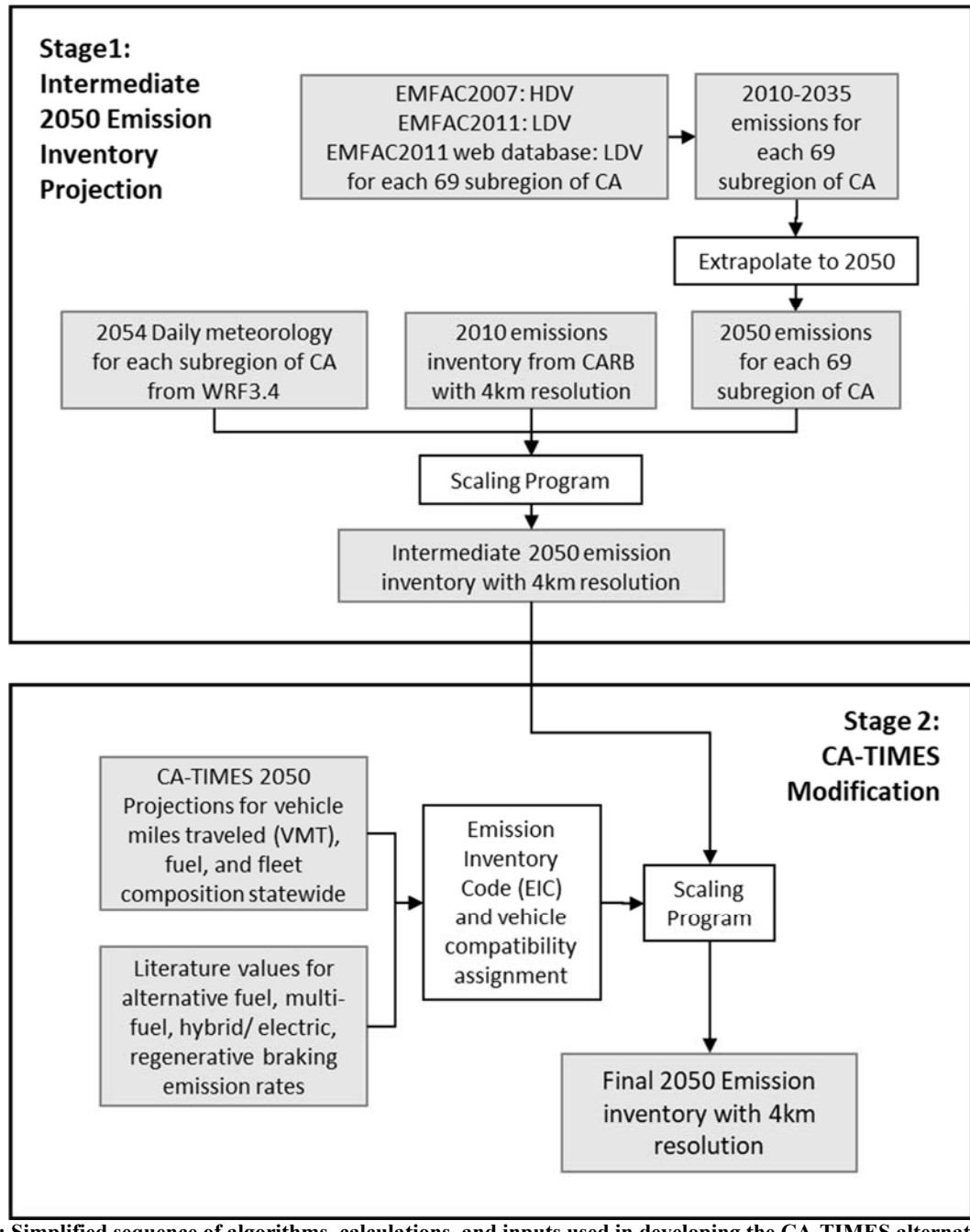

**Figure 2: Simplified sequence of algorithms, calculations, and inputs used in developing the CA-TIMES alternative fuel**
**on-road mobile emissions inventory per scenario. EIC is emission inventory code.**

### 2.2.1 EMFAC Emissions and Activity Projections

Criteria pollutant emissions for on-road mobile sources in future years were forecast using the EMFAC 2011 model developed by the California Air Resources Board (CARB) (California Air Resources Board 2011). EMFAC 2011 accounts for annual VMT trends and vehicle fleet composition turnover using Department of Motor Vehicle (DMV) data. EMFAC incorporates the latest on-road mobile policies including the Low Emission Vehicle emission standards, Low Carbon Fuel Standard (LCFS), Pavley Clean Car Standard, and the Truck and Bus ruling (California Air Resources Board, 2011). EMFAC 2011 predicts past, present, and future year (up to 2035 or 2040) emissions including anticipated future emissions standards and regulations specific to California. EMFAC predicts emissions and energy activity (VMT, trips, vehicles, gallons fuels) for 69 Geographical Area Indexes (GAIs) which represent the intersection of air basins and counties (listed in Table S1).

In the current study, EMFAC was run for each calendar year from 2020–2035 to infer the emissions trends that could then be extrapolated to 2050. A simple linear regression model was used to represent VMT over the period 2020-2035, while a logarithmic regression model was fit to pollutant emissions for each vehicle type over the same time period. Future studies will use EMFAC 2014 which directly predicts emissions in 2050 making this step unnecessary.

### 2.2.2 Spatial Allocation of Mobile Source Emissions in an Intermediate 2050 Inventory

An existing on-road mobile emissions inventory for the year 2010 with 4 km spatial resolution served as the starting point for the projection of an intermediate emissions inventory in 2050. Scaling factors to account for VMT growth and adoption of existing policies were first calculated as the ratios between EMFAC emissions from 2010 and (extrapolated) 2050 within each of the 69 GAI regions. Separate scaling factors were developed for each pollutant emitted from different vehicle classes and control technologies as represented by unique emission inventory codes (EICs). The combined intermediate emissions (em) scaling factor $SF_{act + met}$ defined in equation (3) reflects independent changes in activity (act) (Eq. 1) and meteorology (met) (Eq. 2). Future 2054 temperature and relative humidity generated at 4km resolution with WRF3.2 (Zhang, Chen et al. 2014) were averaged to GAI regions used by EMFAC to produce hour-specific reactive organic gas (ROG) emission rates that vary from the annual average emission rates. Activity is either defined as vehicle miles travelled (VMT) or vehicle trips, depending on the emission process. For example, activity equals VMT for tailpipe emission rates (e.g. grams NO mile$^{-1}$) or tire and brake wear emissions (grams PM mile$^{-1}$). Otherwise, activity equals the number of vehicles within each type/fuel/aftertreatment category such as for evaporative emissions of non-methane hydrocarbons (grams NMHC vehicle$^{-1}$) from the fuel system (non-tailpipe emissions). Emission rates are highly dependent on the emission process (evaporative, exhaust, tire or brake wear), fuel (gasoline or diesel) and the aftertreatment device (catalytic or non-catalytic).

Emissions within each 4km grid cell of the 2010 inventory are multiplied by the 2050 to 2010 scaling factor $SF_{act+met}$ to estimate the "intermediate" 2050 emissions that will be further modified according to various additional policy choices represented in CA-TIMES.

$$SF_{act} = \frac{em(act_{2050}, met_{2010})}{em(act_{2010}, met_{2010})} \tag{1}$$

$$SF_{met} = \frac{em(act_{2010}, met_{2050})}{em(act_{2010}, met_{2010})} \tag{2}$$

$$SF_{act+met} = SF_{act} \cdot SF_{met} \tag{3}$$

### 2.2.3 CA-TIMES Modification of Intermediate 2050 On-Road Mobile Emissions

State-wide CA-TIMES scaling factors were applied uniformly at all locations to the 2050 intermediate emissions inventory described in the previous section to produce the final 2050 emissions inventory. EMFAC accounts for population growth and emissions changes that are required by existing air quality rules and regulations through 2050. CA-TIMES accounts for additional changes that will be required to comply with state GHG targets but which have not yet been placed into emissions rules and regulations. The final inventory retains the spatial and temporal features inherent in the intermediate emissions inventory but incorporates updated information about new fuels, technologies, and emissions rates based on state-wide predictions from CA-TIMES (Fig. 3).

EMFAC vehicles classes expressed as EIC codes were mapped to compatible vehicle classes used by CA-TIMES as described in Table S2. Spark ignition (gasoline) vehicles in CA-TIMES were further classified as catalyst-equipped or non-catalyst-equipped to match EMFAC categories. EMFAC resolves non-catalyst-equipped and catalyst-equipped gasoline vehicles into several sub-categories (light-heavy duty truck (LHDT) and heavy-heavy duty truck (HHDT) (see Table S2 for complete description of vehicle classes) while CA-TIMES does not include this level of resolution.

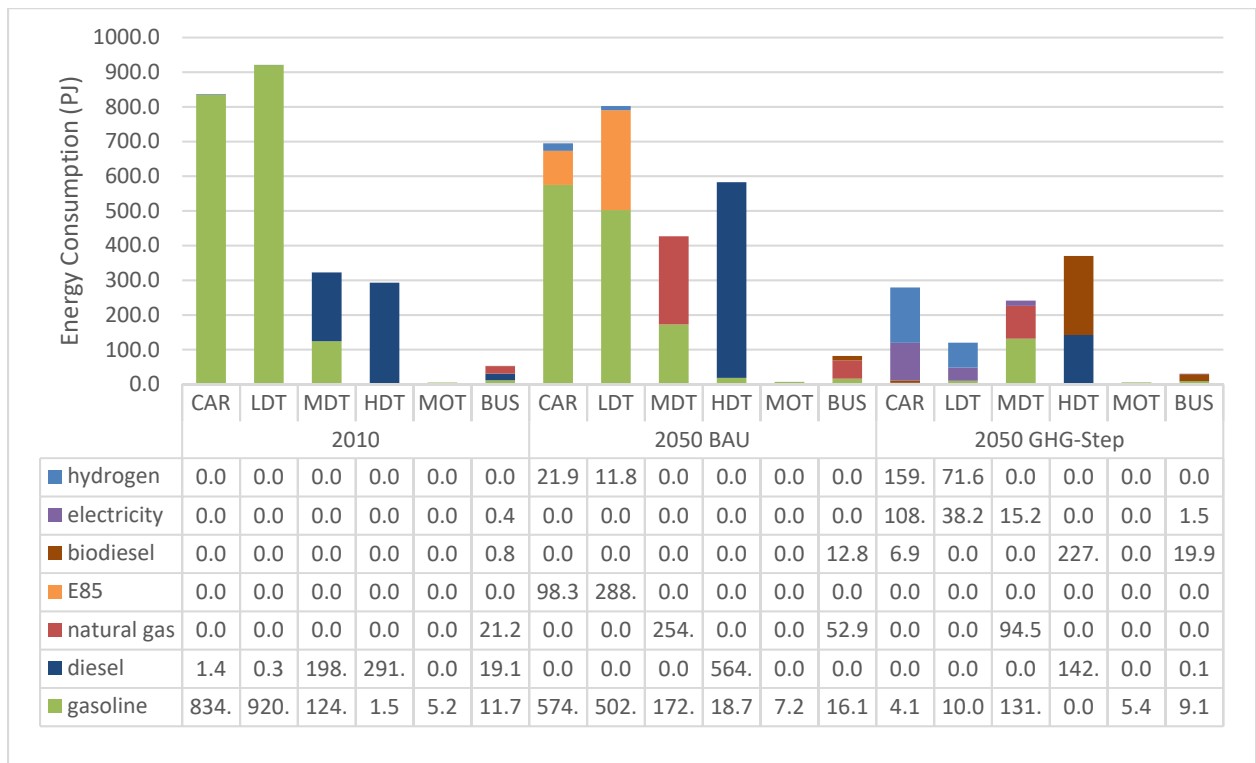

| | CAR | LDT | MDT | HDT | MOT | BUS | CAR | LDT | MDT | HDT | MOT | BUS | CAR | LDT | MDT | HDT | MOT | BUS |
|---|---|---|---|---|---|---|---|---|---|---|---|---|---|---|---|---|---|---|
| | | | 2010 | | | | | | 2050 BAU | | | | | | 2050 GHG-Step | | | |
| hydrogen | 0.0 | 0.0 | 0.0 | 0.0 | 0.0 | 0.0 | 21.9 | 11.8 | 0.0 | 0.0 | 0.0 | 0.0 | 159. | 71.6 | 0.0 | 0.0 | 0.0 | 0.0 |
| electricity | 0.0 | 0.0 | 0.0 | 0.0 | 0.0 | 0.4 | 0.0 | 0.0 | 0.0 | 0.0 | 0.0 | 0.0 | 108. | 38.2 | 15.2 | 0.0 | 0.0 | 1.5 |
| biodiesel | 0.0 | 0.0 | 0.0 | 0.0 | 0.0 | 0.8 | 0.0 | 0.0 | 0.0 | 0.0 | 0.0 | 12.8 | 6.9 | 0.0 | 0.0 | 227. | 0.0 | 19.9 |
| E85 | 0.0 | 0.0 | 0.0 | 0.0 | 0.0 | 0.0 | 98.3 | 288. | 0.0 | 0.0 | 0.0 | 0.0 | 0.0 | 0.0 | 0.0 | 0.0 | 0.0 | 0.0 |
| natural gas | 0.0 | 0.0 | 0.0 | 0.0 | 0.0 | 21.2 | 0.0 | 0.0 | 254. | 0.0 | 0.0 | 52.9 | 0.0 | 0.0 | 94.5 | 0.0 | 0.0 | 0.0 |
| diesel | 1.4 | 0.3 | 198. | 291. | 0.0 | 19.1 | 0.0 | 0.0 | 0.0 | 564. | 0.0 | 0.0 | 0.0 | 0.0 | 0.0 | 142. | 0.0 | 0.1 |
| gasoline | 834. | 920. | 124. | 1.5 | 5.2 | 11.7 | 574. | 502. | 172. | 18.7 | 7.2 | 16.1 | 4.1 | 10.0 | 131. | 0.0 | 5.4 | 9.1 |

**Figure 3: CA-TIMES' energy consumption by vehicle weight class, fuel, and scenario for on-road sources. Vehicle categories include car, light duty truck (LDT), medium duty truck (MDT), heavy duty truck (HDT), motocycles (MOT), and bus.**

The use of new fuels in the on-road fleet required special consideration during preparation of the 2050 emissions
inventory.  As a starting point, emission rates from EICs representing conventionally-fueled vehicles were calculated
from 2050 EMFAC output by dividing each pollutant emission by the respective vehicle activity indicator (either
VMT, vehicle number, or fuel consumption) to serve as a baseline for CA-TIMES scenario adjustments. Next, the 181
combinations of alternative fuels and electric hybrid, dedicated or single/multi-fueled applications and vehicles weight
classes were mapped to EMFAC by vehicle class and reference fuel (see Table S2 and S3).  CA-TIMES predicts the
amount of alternative fuel consumed, not the VMT associated with that alternative fuel.  The VMT associated with
each alternative fuel was therefore estimated as the VMT associated with the conventional fuel divided by the energy
content of the consumed conventional fuel ($E_v$) multiplied by the energy content of the alternative fuel ($E_{v,f}$) output
by CA-TIMES.  This calculation assumes that vehicle weight and aerodynamics do not change significantly as
alternative fuels are adopted.  Finally, the emissions rate for each alternative fuel was estimated based on a literature
review of emissions factors for conventional vs. alternative fueled vehicles.  Reference emission rates ($er_{v,ref}$) and
"alternative to conventional" scaling factors ($er_{v,f}$ / $er_{v,ref}$) for the vehicle fuels of interest are listed in Table 1.
**Table 1: Emission rate changes for alternative fuels in on-road vehicles.  Alternative fuels include 85% ethanol 15%**
**gasoline mixture (E85), biodiesel (B100), and compressed natural gas.  Conventional fuels include gasoline, diesel, or ultra**
**low sulfur diesel (USLD).  After treatment devices include three way catalyst (TWC), diesel oxidation catalyst (DOC),**
**diesel particle filter (DPF), exhaust gas recirculation (EGR), and selective catalytic reduction (SCR).**

| Alternative Fuel | Reference Conventional Fuel | After-treatment | Pollutant | Alt/ Conv Ratio | Conv % Change | Data Source |
|---|---|---|---|---|---|---|
| E85 | Gasoline | same (TWC) | CO | 1.00 | 0.0% | Graham, Belisle et al. (2008) |
| | | | NOx | 0.55 | -45% | Graham, Belisle et al. (2008) |
| | | | SOx | 1.00 | 0.0% | Assumed |
| | | | ROG | 1.00 | 0.0% | Graham, Belisle et al. (2008) |
| | | | PM | 0.25 | -75% | Hays, Preston et al. (2013) |
| B100 | Diesel or ULSD | DOC+ DPF+ EGR+ SCR | CO | 0.03 | -97% | Alleman, Eudy et al. (2004), Alleman, Barnitt et al. (2005), Hasegawa, Sakurai et al. (2007) |
| | | | NOx | 0.85 | -15% | Alleman, Eudy et al. (2004), Alleman, Barnitt et al. (2005), Tsujimura, Goto et al. (2007) |
| | | | SOx | 1.00 | 0.0% | Assumed |
| | | | ROG | 0.03 | -97% | Alleman, Eudy et al. (2004), Alleman, Barnitt et al. (2005), Hasegawa, Sakurai et al. (2007) |
| | | | PM | 0.03 | -97% | Alleman, Eudy et al. (2004), Alleman, Barnitt et al. (2005), Hasegawa, Sakurai et al. (2007), Rounce, Tsolakis et al. (2012) |
| CNG | Diesel or ULSD | TWC | CO | 0.67 | -33% | Cooper, Arioli et al. (2012) |
| | | | NOx | 0.19 | -81% | Cooper, Arioli et al. (2012) |
| | | | SOx | 1.00 | 0.0% | Assumed |
| | | | ROG | 0.34 | -66% | Cooper, Arioli et al. (2012) |
| | | | PM | 0.08 | -92% | Cooper, Arioli et al. (2012) |


Equation (4) illustrates how the total emissions ($em_v$) were calculated for a given vehicle class (subscript v) by
summing the product of the emission rate and VMT for each fuel (subscript f) for the number of different fuels (n)
consumed by that vehicle as defined by each CA-TIMES scenario.
$$em_v = \sum_f^n er_{v,ref} \cdot \underbrace{\frac{er_{v,f}}{er_{v,ref}}}_{\substack{\text{Alternative} \\ \text{fuel/energy} \\ \text{emission} \\ \text{rate}}} \cdot act_v \cdot \underbrace{\frac{E_{v,f}}{E_v}}_{\substack{\text{Proportion of} \\ \text{activity by fuel/} \\ \text{energy for} \\ \text{vehicle}}} \qquad (4)$$
where

v = vehicle type by weight

f = unconventional or alternative fuel type from f1, f2, f3…n

ref = reference (conventional) fuel, typically gasoline or diesel.

$em_v$ = emissions for a give vehicle type per pollutant. Where pollutant is ROGs, CO, NOx, PM10, SOx

[tons pollutant].

$er_{v,ref}$ = pollutant emission rate for a vehicle using the reference (conventional) fuel based from EMFAC

[tons pollutant VMT$^{-1}$ or tons pollutant vehicle$^{-1}$]

$er_{v,f}$ = pollutant emission rate for a vehicle using an alternative fuel based from EMFAC [tons pollutant

VMT$^{-1}$ or tons pollutant vehicle$^{-1}$]

$act_v$ = total vehicular activity (not divided by fuel) [VMT or vehicles]

$e_{v,f}$ = energy consumption for a given fuel by vehicle given by CA-TIMES scenario [PJ]

$e_v$ = total energy consumed for vehicle for all fuels by CA-TIMES scenario [PJ]


Alternative fuels considered by CA-TIMES include 95% volume blend methanol (M95), 85% volume blend ethanol
(E85), compressed natural gas (CNG), liquid petroleum gas (LPG), biodiesel, compressed or liquid hydrogen, and
electric drivetrains.  Electric vehicles (EVs) include hybrid, (HEV), plug-in hybrid (PHEV), and plug-in or battery
(PEV or BEV).  CA-TIMES often predicted the use of multiple technologies and fuels within the same vehicle
weight class (see Table S4 through Table S12 for complete lists).  For example, in the case of a hybrid diesel electric
vehicle which runs on 3 energy sources, diesel, biodiesel, and electricity, (e.g. a biodiesel PHEV MDT), 3 sets of
emission rates (1 for each fuel) were estimated to replace the single emissions rate for the traditional CI engine for
this vehicle class (diesel MDT).

Only approximately 10% of the possible vehicle type/fuel/engine combinations considered by CA-TIMES (see
Table S4 to Table S12) were actually used in the 2050 BAU and GHG-Step scenarios as the model optimized for
low cost and low-carbon solutions.  The main alternative liquid or gaseous fuels projected by CA-TIMES were E85,
biodiesel, and CNG.  CA-TIMES predicted that E85 would displace gasoline while biodiesel and CNG would
displace diesel based on the dominant fuel consumed for the same vehicle weight class counterpart.  This fuel
substitution alters emissions rates for criteria pollutants as shown in Table 1. For battery electric or fuel cell
vehicles, the conventional fuel displaced was based on the dominant fuel for that vehicle class, e.g. gasoline for
LDVs.

**2.2.4 On-Road Mobile PM and Gas Speciation and Size Profile Changes**
Tailpipe exhaust, fuel tank evaporative, and brake wear emissions were adjusted when the vehicle fuel or technology
was changed.  This requires new source profiles to be defined for E85, biodiesel, and CNG fueled vehicles to describe
their emissions of speciated volatile organic compounds (VOCs) and size & composition-resolved particulate matter.
New emissions inventory codes (EICs) were created (summarized in Table S13) and associated with new VOC and
PM emissions profiles (summarized in Tables S14 – S16) for this purpose**.**
Multiple measurements are available in the literature for the composition of exhaust from ethanol-fueled vehicles.  In
the present study, the average VOC profiles measured using the Federal Test Procedure (FTP), Unified Cycle (UC),
and US06 high speed drive cycles were used for the hot running E85 VOC exhaust (Haskew and Liberty 2011).  The
FTP phase 1 profile was applied for the cold-start E85 VOC emissions (Haskew and Liberty 2011).  E85 PM size
distributions are summarized in Table S15 (Szybist, Youngquist et al. 2011), while PM composition information is
summarized in Table S16 (Ferreira da Silva, Vicente de Assuncao et al. 2010, Hays, Preston et al. 2013)**.**  Figure 4
illustrates the size and composition distribution of particulate matter emitted from catalyst-equipped gasoline vehicles
and catalyst-equipped vehicles fueled by 85% ethanol and 15% gasoline (E85) as an example.

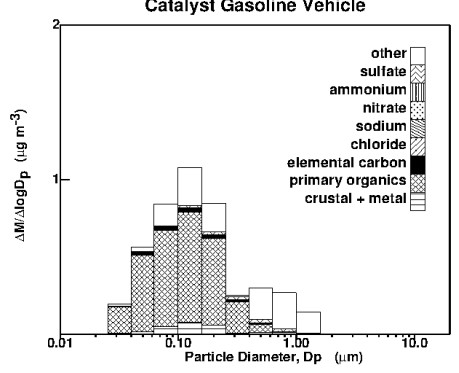 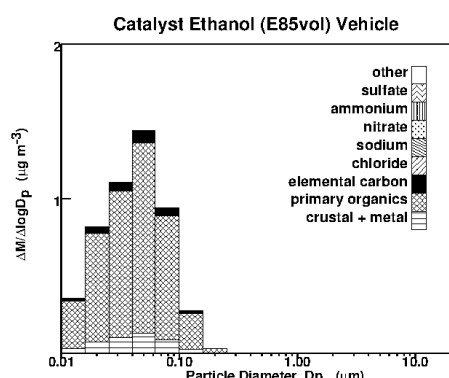


**Figure 4: Particle emissions size and composition distribution for catalyst equipped gasoline vehicles (left panel) and**
**catalyst equipped ethanol (E85) vehicles (right panel).**
Aftertreatment devices were found to be more influential on biofuel exhaust rates (Alleman, Eudy et al. 2004, Alleman,
Barnitt et al. 2005, Frank, Tang et al. 2007, Hasegawa, Sakurai et al. 2007, Tsujimura, Goto et al. 2007, Rounce,
Tsolakis et al. 2012) than changes to fuel properties and feedstock origin (Graboski, McCormick et al. 2003, Durbin,
Cocker et al. 2007).  Diesel particulate filters (DPF), exhaust gas recirculation (EGR), selective catalytic reduction
(SCR), and oxidation catalyst (OC) were assumed to be deployed on diesel and biodiesel powered vehicles by 2050.
PM size distributions for DPF-equipped vehicles were obtained from (Rounce, Tsolakis et al. 2012) (Table S15), and
trace element, carbonaceous and inorganic ion fractions of PM distributions were obtained from (Cheung, Polidori et
al. 2009, Cheung, Ntziachristos et al. 2010) (see Table S16). Gas-phase VOC emissions profiles for biodiesel were
not updated from fossil diesel profiles in the current study, but this change will be considered in future work.
The CNG VOC profile and PM size distribution was constructed based on (Gautam 2011) (Tables S14 and S15). PM
emissions of carbonaceous compounds, metals, and ions were measured from CNG vehicles running on the UDDS
driving cycle (Yoon, Hu et al. 2014) (see Table S16). Figure 5 illustrates the size and composition distribution of
particulate matter emitted from diesel vehicles, bio-diesel vehicles equipped with a diesel particle filter and exhaust
gas recirculation, and catalyst-equipped CNG vehicles.

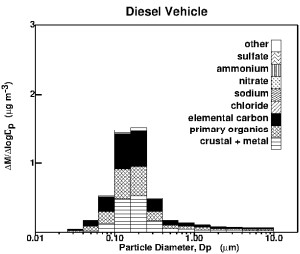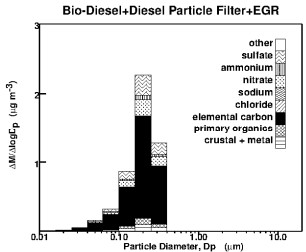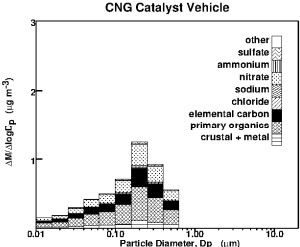


**Figure 5: Particle emissions size and composition distribution for diesel vehicles (left panel), bio-diesel vehicles (center**
**panel), and CNG catalyst equipped vehicles (right panel).**
All fully electric vehicles, such as battery electric vehicles (BEVs) and H2 fuel cell vehicles, were assumed to have
zero tailpipe exhaust and evaporative emission rates. Brake wear emission rates were reduced by 59% (Antanaitis
2010) for all partial or fully electric vehicles equipped with regenerative breaking, such as hybrid, electric battery or
fuel cell vehicles. Tire wear emissions were assumed to be independent of fuel or technology type.
**2.3 CA-REMARQUE Aviation, Rail, and Off-Road Algorithms**
Aviation sources include commercial, civil, agricultural, or military use and primarily run on jet fuel or aviation
gasoline. The rail emission sources include passenger, commuter, switching and hauling trains which currently run
primarily on diesel fueled generators powering an electric drivetrain. Off-road equipment includes industrial,
agricultural, and construction equipment, port and rail operations, as well as lawn and garden equipment. The list of
aviation, rail, and off-road emission source categorizations are based on the EICs listed in Table S17 (including new
EICs created to represent sources operating on alternative fuels previously not in the CARB inventory).
**2.3.1 VISION Model**
Future 2050 emissions for aviation, rail, and off-road equipment were assumed to follow the 2010 versus 2050 growth
projected by the CARB VISION model (California Air Resources Board 2012), an off-road expansion of Argonne's
on-road VISION model (Argonne National Laboratory Transportation Technology R&D Center 2012). CARB's off-
road VISION model uses historical trends to project to the year 2050 while incorporating some future standards for
criteria pollutant emission rates. These include the implementation of Tier 4 130-560 kW compression-ignition diesel
engine emission standards for PM, CO, and NMHC+NOx (California Air Resources Board 2010) leading to 90%
reduction in PM emissions rates and an 85% reduction in NMHC and NOx emissions rates.
Aviation, rail, and off-road 2010 emissions at 4 km resolution ($em^{2010}_{cell,I}$) were scaled to produce an "intermediate"
estimate prior to CA-TIMES adjustments using Eq. (5).

$$em^{2050}_{cell,i,intermediate} = \underbrace{\left(\frac{em^{2050}_i}{em^{2010}_i}\right)}_{\substack{\text{State-wide} \\ \text{emission growth} \\ \text{scaling from 2010} \\ \text{to 2050}}} \cdot em^{2010}_{cell,i} \tag{5}$$
where
$em^{2050}_{cell,i,intermediate}$ = intermediate grid cell 2050 emissions for a transport source (aviation, rail, off-road)
consuming a reference or conventional fuel or energy [kg hr$^{-1}$]
$em^{2050}_i$ = state-wide 2050 emissions of a transport source [kg hr$^{-1}$ or tons day$^{-1}$]
$em^{2010}_i$ = state-wide 2010 emissions of a transport source [kg hr$^{-1}$ or tons day$^{-1}$]
$em^{2010}_{cell,i}$ = grid cell 2010 emissions of a transport source [kg hr$^{-1}$]

**2.3.2 CA-TIMES Modification of Intermediate 2050 Off-Road Mobile Emissions**
The portion of energy consumed for each fuel ($E_{i,f}/\Sigma f\ E_{i,f}$) as projected by CA−TIMES was applied to the
intermediate 2050 emissions inventory for each transport mode (f) and source type (i) using Eq. (6). The
consumption of different fuels relative to total fuel consumption for a given mode is shown in Fig. S1-S3 for rail,
off−road, and aviation modes respectively. Alternative to conventional scaling factors were applied to account for
adoption of alternative fuels as summarized in Table 2. Eq. (6) also includes an after treatment or control device
factor (1−η) where appropriate.
$$SFi,f = \underbrace{\left(\frac{E_{i,f}}{\Sigma_f E_{i,f}}\right)}_{\substack{\text{Portion of} \\ \text{alternative} \\ \text{fuel energy} \\ \text{consumption}}} \cdot \underbrace{\left(\frac{em_{i,f}^{2050}}{em_{i,intermediate}^{2050}}\right)}_{\substack{\text{Alternative} \\ \text{fuel} \\ \text{emission} \\ \text{scaling} \\ \text{relative to} \\ \text{conventional}}} \cdot \underbrace{(1 - \eta_i)}_{\substack{\text{Fraction of} \\ \text{pollutant not} \\ \text{removed by} \\ \text{aftertreatment} \\ \text{device}}} \tag{6}$$
where
$SF_{i,f}$ = emission scaling factor for a given new/alternative or non-conventional/non-reference fuel for a
transport source [dimensonless]
$E_{i,f}$ = new/alternative fuel/energy consumed by a transport source (e.g. biodiesel for commuter rail) [PJ]

$\sum_f E_{i,f}$ = total fuel/energy consumed by a transport source (e.g. biodiesel + diesel for commuter rail) [PJ]

$em_{i,f}^{2050}$ = state-wide 2050 emissions of a transport source consuming a new/alternative fuel [kg hr$^{-1}$ or
tons day$^{-1}$]

$em_{i,intermediate}^{2050}$ = state-wide 2050 intermediate emissions of a transport source consuming a
new/alternative fuel. [kg hr$^{-1}$ or tons day$^{-1}$]

$\eta_i$ = efficiency of removal from a control or aftertreatment device [fraction from 0.00-1.00]

**Table 2: Emission rate changes for alternative fuels in off-road vehicles.**

| Transport Mode | Alternative Fuel | Reference Conven-tional Fuel | Pollutant | Alt/ Conv Ratio | Conv % Change | Citations |
|---|---|---|---|---|---|---|
| **Rail** | Biodiesel | Diesel | CO | 0.655 | -34.5% | Osborne, Fritz et al. (2010) |
| | | | NOx | 1.13 | 13% | Osborne, Fritz et al. (2010) |
| | | | SOx | 0.0005 | -99.95% | Assumed (see text) |
| | | | ROG | 0.775 | -22.5% | Osborne, Fritz et al. (2010) |
| | | | PM | 0.805 | -19.5% | Osborne, Fritz et al. (2010) |
| **Off-road/ Agricultural** | Biodiesel | Diesel | CO | 1 | 0% | Durbin, Cocker et al. (2007) |
| | | | NOx | 1.08 | 8% | Durbin, Cocker et al. (2007) |
| | | | SOx | 1 | 0% | Durbin, Cocker et al. (2007) |
| | | | ROG | 0.39 | -61% | Assumed (see text) |
| | | | PM | 1.13 | 13% | Durbin, Cocker et al. (2007) |
| | Compressed natural gas | Diesel | CO | 0.668 | -33.2% | Cooper, Arioli et al. (2012) |
| | | | NOx | 0.189 | -81.1% | Cooper, Arioli et al. (2012) |
| | | | SOx | 1 | 0% | Assumed (see text) |
| | | | ROG | 2.349 | 134.9% | Cooper, Arioli et al. (2012) |
| | | | PM | 0.0782 | -92.18% | Cooper, Arioli et al. (2012) |
| **Aviation** | Biomass-based kerosene jet fuel | Kerosene jet fuel | CO | 1 | 0% | Lobo, Rye et al. (2012) |
| | | | NOx | 1 | 0% | Lobo, Rye et al. (2012) |
| | | | SOx | 0.007 | -99.3% | Assumed (see text) |
| | | | ROG | 0.605 | -39.5% | Lobo, Rye et al. (2012) |
| | | | PM | 0.38 | -62% | Lobo, Hagen et al. (2011) |

The final emissions for each specific offroad source consuming each specific fuel in 2050 ($em_{cell,i,f}^{2050}$) are then
calculated by combining the effects of the VISION and CA-TIMES updates as shown in Eq. (7).
$$em_{cell,i,f}^{2050} = SF_{i,f} \cdot em_{cell,i,intermediate}^{2050} \tag{7}$$
Aviation biomass-based kerosene jet fuel (KJF) emissions changes are based on Fischer-Tropsch gas-to-liquid (FT
GTL) biofuel aviation emissions tests (Lobo, Hagen et al. 2011, Lobo, Rye et al. 2012). These studies found minor
changes to CO and NO$_x$ emissions due to the adoption of biofuels. SO$_x$ reduction was assumed proportional to the
fuel sulfur content (Lobo, Rye et al. 2012) leading to reductions of 99% as shown in Table 2.
Off-road equipment (other than trains) operating on biodiesel instead of Ultra low-sulfur diesel (ULSD) was assumed
to emit HC and $NO_x$ with scaling factors (relative to conventional diesel emissions) of 0.39 and 1.08, respectively
(Durbin, Cocker et al. 2007). No significant changes in CO, $SO_x$ and PM due to the adoption of biodiesel vs. ULSD
were identified in the literature and so these emissions were assumed to remain at levels estimated for conventional
diesel engines. This approach inherently assumes that the sulfur content of biodiesel will not exceed the current limit
of 15 ppm for ULSD. Off-road or agricultural emission changes from switching from diesel to CNG are also found
to have large reductions in most pollutants except reactive organic gases (ROGs) (Cooper, Arioli et al. 2012).
Military aviation emissions were held constant at 2010 levels in the current study due to an assumption of continued
exemptions for military activity.
**2.3.3 Off-Road Mobile PM and Gas Speciation and Size Profile Changes**
PM mass size distributions for E85, biodiesel, and CNG are assumed to be similar for off-road and on-road vehicles
(Table S15)**.** The new PM mass size distribution for biomass-based KJF is shown in Table S18 (Lobo, Hagen et al.
2011). Figure 6 illustrates the size and composition distribution of particulate matter emitted from conventional jet-
fuel aircraft and biomass-based kerosene jet fuel aircraft. The conventional profile is based on old source profile
measurements that assumed uniform distribution of particles between diameters 0.1-1.0 μm. This conventional profile
will be updated with more recent literature values in future work.

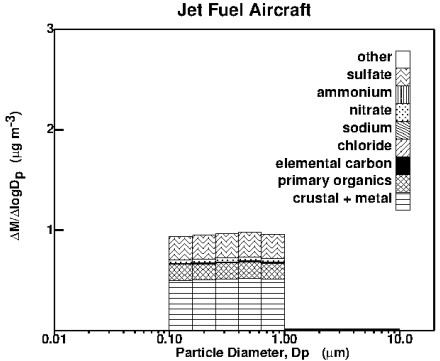 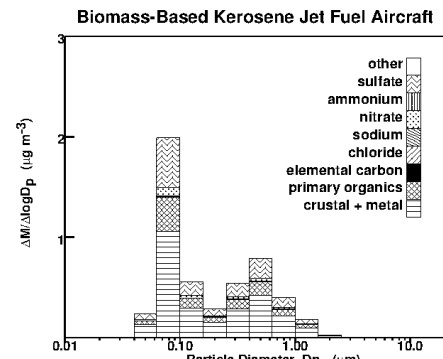


**Figure 6: Particle emissions size and composition distribution for jet-fueled aircraft (left panel) and biomass-based**
**kerosene jet-fueled aircraft (right panel).**
**2.4 CA-REMARQUE Marine Algorithms**
The marine emission source category includes all ocean going vessels (OGV), commercial harbor craft (CHC), and
recreational boats (see Table S19). An intermediate OGV emissions inventory was predicted for the year 2050 based
on the extrapolation of Port of Los Angeles and Port of Long Beach 2020 trends (Starcrest Consulting Group 2009,
The Port of Los Angeles and The Port of Long Beach 2010) (see Table S20). All other OGV emissions (not listed in
Table S20) in California were held constant at 2010 levels in the intermediate 2050 inventory prior to modifications
from CA-TIMES.

### 2.4.1 CA-TIMES Modification of Intermediate 2050 Marine Emissions

The fuels used to power OGVs were modified based on predictions from the CA-TIMES' scenarios. It should be noted that the CA-TIMES model reports worldwide marine energy consumption. In the current study, it was assumed that marine vessels operating near the California coast would consume the global average mix of biofuels produced by CA-TIMES. For example, if CA-TIMES indicates that a third of the residual fuel oil (RFO) (also call heavy fuel oil) consumed globally by marine vessels would be converted to biomass-based residual fuel oil (BRFO), then a third of the RFO marine vessel emissions near California boundaries were also converted to BRFO. As indicated by Fig. S4, CA-TIMES finds other approaches besides biofuel adoption for ships are more cost-effective for meeting the GHG target in 2050. CA-TIMES determined that it will be more economical to substitute some RFO with a lighter petroleum (diesel) to decrease carbon intensity rather than using biomass-based RFO.

Alternative fuels used in marine sources will modify criteria pollutant emissions. Biomass-based alternatives for marine residual fuel oil (RFO) were estimated to be similar to the average of B100 from palm oil, animal fat, soybean oil, and sunflower oil operating at 75% load (Petzold, Lauer et al. 2011). $NO_x$ was the only regulated pollutant observed to remain constant during emissions testing. Emissions of all other pollutants decreased as summarized in Table 3.

Table 3: Emission rate changes from ships changing from conventional fuels to biofuels.

| Alternative Fuel | Reference Conventional Fuel | Pollutant | Alt/ Conv Ratio | Conv % Change | Citations |
|---|---|---|---|---|---|
| biomass-based residual fuel oil (RFO) | residual fuel oil (RFO) | CO | 0.697 | -30.3% | (Petzold, Lauer et al. 2011) |
| | | NOx | 1 | 0% | (Petzold, Lauer et al. 2011) |
| | | SOx | 0.012 | -98.8% | (Petzold, Lauer et al. 2011) |
| | | ROG | 0.413 | -58.7% | (Petzold, Lauer et al. 2011) |
| | | PM | 0.223 | -77.7% | (Petzold, Lauer et al. 2011) |
| Biodiesel (BDL) | Diesel (DSL) | CO | 0.921 | -7.9% | (Jayaram, Agrawal et al. 2011) |
| | | NOx | 1 | 0% | (Jayaram, Agrawal et al. 2011) |
| | | SOx | 0.0003 | -99.97% | Assumed (see text). |
| | | ROG | 1 | 0% | (Jayaram, Agrawal et al. 2011) |
| | | PM | 0.684 | -31.6% | (Jayaram, Agrawal et al. 2011) |

Assuming biodiesel (BDL) and biomass based residual fuel oil (BRFO) has about 1 ppm sulfur content, and that by 2010 the sulfur content regulations ensured that marine diesel oil (MDO) and RFO had 1.5 ppm and 2.5 ppm S, respectively, then the switch to biofuels would reduce $SO_x$ emissions by 33.3% (relative to conventional MDO) and 60% (relative to conventional RFO). Additional reductions in CO, TOG, and PM were also projected based on (Jayaram, Agrawal et al. 2011, Petzold, Lauer et al. 2011) as summarized in Table 3.

Several international and California shoreline regulations were applied to marine emissions in the year 2050 as summarized in Table S21 and Table S22. At-berth or hotelling container, passenger (cruise), and refrigeration OGVs will use shoreline power instead of auxiliary engines for 80% of their berthing hours by 2020, (California Air Resources Board 2007). It was also assumed that MDO or marine gasoline oil (MGO) used within 24 nautical miles of the California shore will have sulfur content of <0.1% by 2050 (California Air Resources Board 2011). Further

offshore, all marine fuels used within 100 nautical miles of North America were assumed to have sulfur content < 1%
after the year 2012 (leading to reductions shown in Table 3).
**2.4.2 Marine PM and Gas Speciation and Size Profile Changes**
PM size distribution changes caused by the switch to alternative marine fuels were based on (Jayaram, Agrawal et al.
2011) (see Table S23). The size and composition distribution profiles used to represent marine emission associated
with different fuels are displayed in Fig. 7.
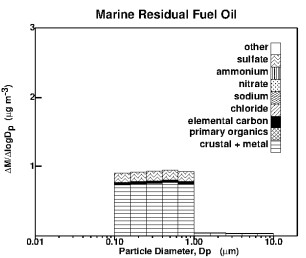 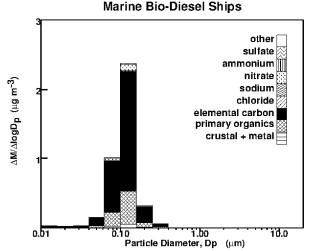 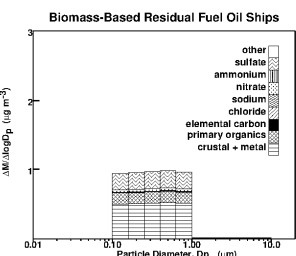

**Figure 7: Particle emissions size and composition distribution for ships powered by marine residual oil (left panel),**
**marine bio-diesel (center panel), and biomass-based residual fuel oil (right panel).**

**2.5 CA-REMARQUE Residential and Commercial Algorithms**
Major emissions sources within the residential and commercial sectors include natural gas combustion (space heating
and water heating), biomass combustion ( fireplaces and stoves), and food cooking (especially charbroiling and
frying). The residential and commercial emissions associated with natural gas and food cooking were assumed to
scale according to population growth projected for each county (Table S24) (State of California 2013) to produce an
intermediate emissions inventory. These intermediate residential and commercial gridded emissions were then scaled
to reflect 2010 versus 2050 results from CA-TIMES (Fig. 8).
Natural gas consumption in the commercial sector reduced by half (325 PJ to 162 PJ) in the GHG-Step scenario
relative to the BAU scenario in 2050. Most of commercial energy reduction is due to efficiency gains and switch
from natural gas to electrification of end uses. Natural gas consumption in the residential sector also decreases (615
PJ to 507 PJ) under the GHG-Step scenario relative to the BAU scenario. Much of the energy that would have been
supplied by natural gas is replaced by renewable sources such as solar (155 PJ) which was assumed to have no criteria
pollutant emissions in California. Improved energy efficiency and conservation also plays a role, with residential
electricity consumption decreasing (402 PJ to 313 PJ) in the GHG-Step scenario. Other combustion sources, including
wood burning and distillate oil fuel consumption, were allowed to compete in CA-TIMES subject to the constraint
that they could not increase above the 2010 levels in order to maintain compliance with current air quality regulations.


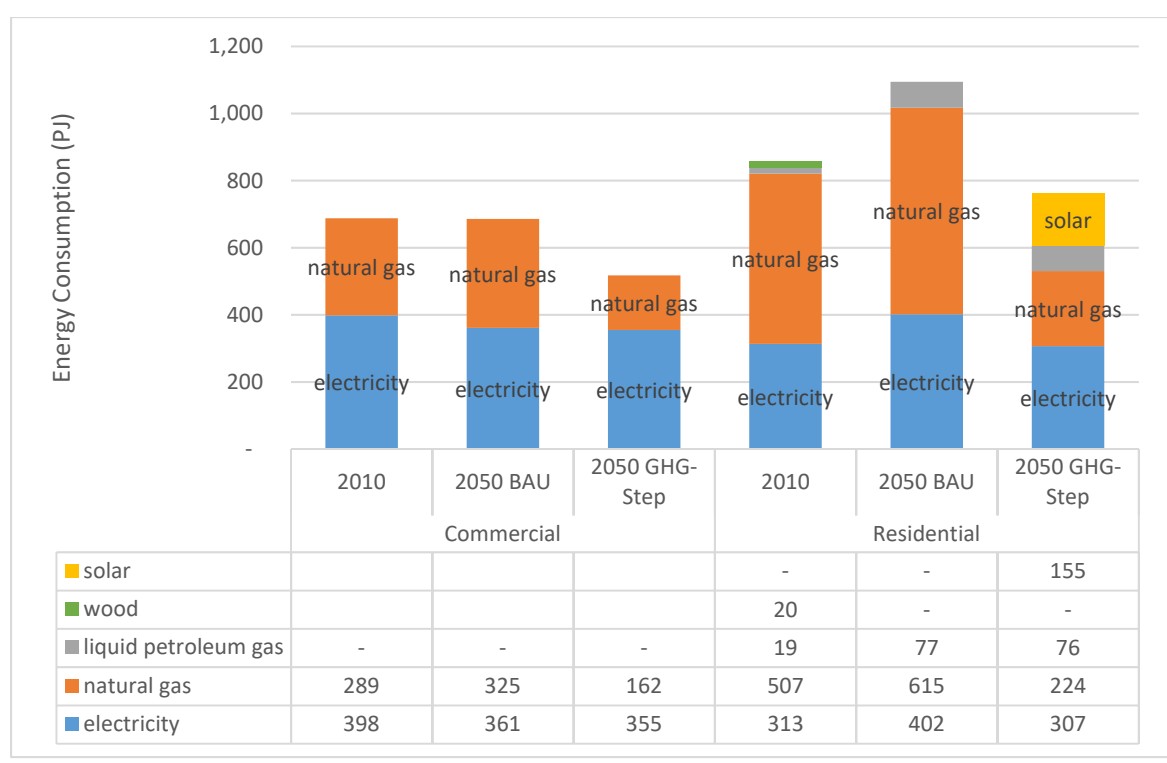

| | 2010 | 2050 BAU | 2050 GHG-Step | 2010 | 2050 BAU | 2050 GHG-Step |
|---|---|---|---|---|---|---|
| | | Commercial | | | Residential | |
| ■ solar | | | | - | - | 155 |
| ■ wood | | | | 20 | - | - |
| ■ liquid petroleum gas | - | - | - | 19 | 77 | 76 |
| ■ natural gas | 289 | 325 | 162 | 507 | 615 | 224 |
| ■ electricity | 398 | 361 | 355 | 313 | 402 | 307 |

**Figure 8: CA-TIMES' energy consumption by energy resource and scenario for commercial and residential.**

**2.6 CA-REMARQUE Electricity Generation Algorithms**

The electricity generation emissions category includes all fuel-burning and renewable power plants for industrial, residential, or commercial use. Annual generation totals for different types of California power plants were extracted from national power plant data (US Energy Information Administration Independent Statistics and Analysis 2012, US Environmental Protection Agency 2014). Emissions rates per unit of fuel burned were estimated for each power plant described in the basecase 2010 emissions inventory.

CA-TIMES finds that non-hydro renewable (geothermal, tidal, solar, wind, and biomass) increases from 10% (22,938 GWh) of the electricity generation mix in 2010 (144,825 GWh) to 35% and 76% (489,493GWh) in the 2050 BAU and 2050 GHG-Step scenario, respectively (see Fig. 9). However, total in-state and out-of-state electricity generation in the GHG-Step scenario is 1/3rd larger than the BAU scenario (416,219 GWh versus 643,373 GWh) to meet the increased demand from sectors such as the on-road vehicles with growing hybridization and electrification needed to meet the 2050 carbon constraint. Statewide scaling factors for electricity generation in the 2050 BAU scenario vs. 2010 and the 2050 GHG-Step scenario vs. 2010 are listed in Table S25.

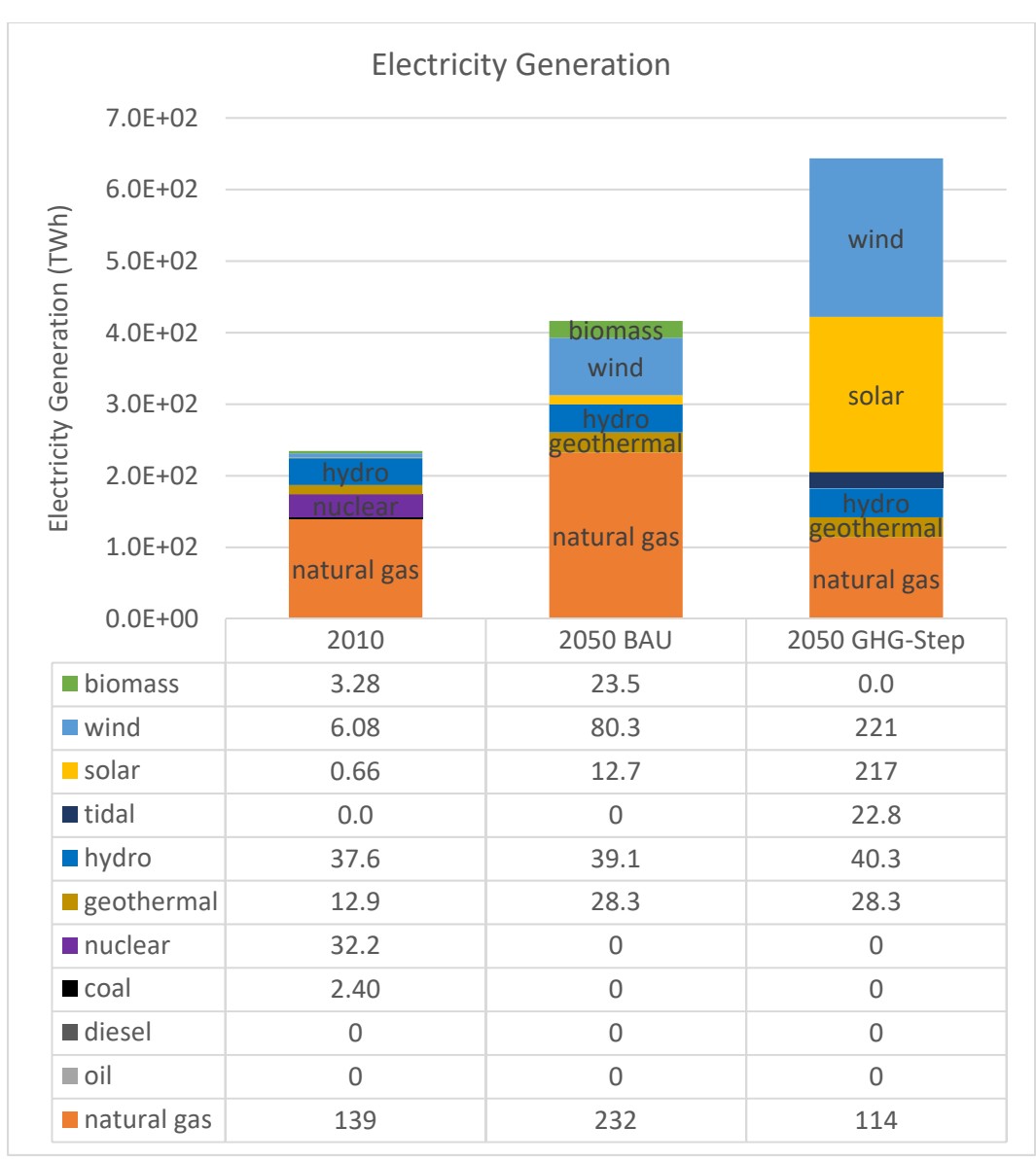

Figure 9: CA-TIMES' electricity generation resource mix by scenario.

CA-TIMES calculates aggregated state-wide energy totals but energy resources (especially for renewables) are not uniformly distributed across the state. In the current study, renewable electricity production in 2050 was spatially allocated in a manner that was consistent with the energy resource potential in 12 regions (Fig. S5) as projected in 15 scenarios by the grid load distribution model SWITCH (Fripp 2012, Johnston, Mileva et al. 2013, Nelson, Mileva et al. 2013). Table S26 lists the electrical generation by energy source for each SWITCH region averaged across these 15 scenarios. This profile of resource potential was then applied to the CA-TIMES predictions summarized in Table S25 yielding the 2050/2010 scaling factors for the BAU scenario (Table S27) and the GHG-Step scenario (Table S28).

The scaling factors summarized in Tables S27 and S28 assume that the out-of-state portion of electricity generation for a given fuel or energy resource in the year 2050 remained constant at 2010 levels. CA-TIMES does not provide additional information describing out-of-state generation except for a few renewables. This out-of-state portion of the

electricity generation was subtracted from the CA-TIMES totals prior to scaling emissions from each power plant in California. Table S29 summarizes the out-of-state portion of electricity generation for each fuel in 2010 and assumed portions in each of the 2050 scenarios.

Additional emissions adjustments were made for new renewable fuels such as those produced by the Biomass Integrated Gasification Combined Cycle (IGCC), a process that gasifies biomass for electricity production. Much of the biomass electricity generation projected by CA-TIMES for 2050 in the BAU scenario uses biomass IGCC (see Tables S30 through S32). There are currently several coal IGCC plants in the US (U. S. Department of Energy National Energy Technology Laboratory 2010, U. S. Department of Energy National Energy Technology Laboratory 2015) but no biomass IGCC plants (Lundqvist 1993, Ståhl and Neergaard 1998, U. S. Department of Energy National Energy Technology Laboratory 2010). Future biomass IGCC emissions in California were estimated using several models that incorporate biomass IGCC, such as GREET, CA-GREET (California Air Resources Board 2009, Argonne National Laboratory Transportation Technology R&D Center 2014, California Air Resources Board 2015), and an NREL analysis (Mann and Spath 1997). Ultimately, biomass IGCC power plant emissions were estimated from conversion of conventional steam turbines in the 2010 ARB inventory based on emissions rates inferred from CA-GREET1.8 for 2050 (Table S33). An inter-comparison study between GREET1.8, GREET 2014, and CA-GREET2.0 showed that theCA-GREET1.8b model had the best agreement with emissions rates from approximately 30 biomass plants operating on wood residue in California.(California Air Resources Board 2011, US Environmental Protection Agency 2014).

## 2.7 CA-REMARQUE Industrial and Agricultural Algorithms

The industrial and agricultural emissions category covers many manufacturing industries such as metal, wood, glass, textile, mining, and chemical. Food and agricultural sectors include farming livestock, crops, food production, bakeries, and breweries. Most of these industries were unchanged in the CA-TIMES energy scenarios, with the notable exception that biofuel and hydrogen fuel production replaced some traditional petroleum production, causing changes in refinery and storage emissions (shown in Figs. S6 to S8).

### 2.7.1 Fossil and Renewable Fuel Production

All fossil petroleum refining and storage emissions in the 2010 ARB emissions inventory were scaled according to the amount of oil production and refining that was required in California for each 2050 CA-TIMES scenario (see Fig. S6). Scaling factors were applied uniformly to all emission processes including seepage, evaporative or fugitive, and other processes. Fossil petroleum consumption generally decreased in future scenarios, but was not eliminated. As discussed in previous sections, transportation modes (e.g. marine, heavy duty trucks) still consume fossil fuel such as diesel, and the stationary sources (electricity generation, residential, and commercial) still consume natural gas. CA-TIMES determined that much of the extracted petroleum used by refineries would be imported to the state rather than extracted locally. This can be seen by the reduction of crude oil supply in California from 1510 PJ in 2010, to 426.5 PJ in the 2050 BAU scenario and 0.0PJ in the GHG-Step scenario (see Fig S6). Refining is also are projected to decline slightly between 2010 and the 2050 scenarios, with reductions of 25% in the BAU scenario and 44% in the

GHG-Step scenario. This suggests that it is more cost effective or less carbon intensive to import fuel than to extract
oil and gas in or around California. The total (imported and in-state) oil supply also decreases in 2050, by -26% in the
BAU (3200PJ) and -44% in the GHG-Step (2400PJ) relative to 2010 (4300PJ). This reflects the adoption of
electrification and alternative fuels to replacing petroleum consumption in the presence of growing energy demand in

481 2050.

Hydrogen ($H_2$) production increased in both 2050 CA-TIMES scenario results, but the increases in the GHG-Step
scenario are much larger (Fig. S7). It was assumed that new hydrogen production facilities would be located at current
$H_2$ production facilities or existing refineries. Overall 32 new natural gas steam methane reforming (SMR) $H_2$ facilities
and 15 new biomass gasification facilities were projected to meet the demand summarized in Fig. S7. In the current
study, criteria pollutant emission rates from SMR $H_2$ production (summarized in Table 4) were calculated from the
top 3 SMR $H_2$ production facilities (California Air Resources Board 2010, California Air Resources Board 2014). Few
studies have been published describing criteria pollutant emissions from biomass gasification $H_2$ production and so
emissions rates for this production pathway were obtained from the CA-GREET model (California Air Resources
Board 2015). Direct criteria pollutant emissions from hydrogen production using electrolysis were zero since this
process uses electricity to split water molecules into $H_2$ and oxygen (emissions from these facilities appear under
electricity generation).
**Table 4: Pollutant emission rate associated with hydrogen production. Unis are grams of pollutant per mmBtu of**
**hydrogen produced.**

|  | SMR - average of top CA H2 SMR facilities | Gasification - CA-GREET2015 Gasification vs. SMR Scaling | Electrolysis |
|---|---|---|---|
| **CO** | 4.303 | 0.997 | 0 |
| **NOx** | 1.701 | 0.34 | 0 |
| **SOx** | 0.092 | 0.406 | 0 |
| **VOC** | 2.33 | 1.118 | 0 |
| **PM10** | 0.433 | 0.048 | 0 |


The CA-TIMES model determined that biofuel consumption and production will be high in California in the year
2050 (Fig. S8). Biofuel refineries for different feedstock classes (wood, municipal solid waste (MSW), herbaceous,
yellow grease or tallow, or corn ethanol) (see Tables S34 and S35) were located using a spatial biomass optimization
model which seeks to minimize cost within resource and regulatory constraints (Tittmann, Parker et al. 2010). Biofuel
refineries were prohibited in NAAQS non-attainment areas, an added constraint based on the high feedstock case
described by (Parker 2012). Production rates at in-state biorefineries were scaled to match the in-state volumes
produced in CA-TIMES for each type of biofuel. Out-of-state imports and refining were assumed for crops that could
not be grown at a large enough scale to meet the demand in California, such as herbaceous crops and the bulk of corn-
ethanol (see Tables S34 and S35). Emissions for each biofuel refinery were estimated using CA-GREET1.8b emission
rates per unit of fuel produced.
**2.7.2 Biogas Capture and Use**
CA-TIMES assumes that landfill gas reduces over time due to better management of organic matter in landfills, and
the consumption of existing landfill stock material over many decades. All biogas in CA-TIMES is converted to
biomethane through removal of $CO_2$ and impurities, and further blended with natural gas so that it is
undistinguishable from extracted fossil natural gas.
Dairy biogas is a significant renewable energy source in CA-TIMES. California produced a fifth of the milk in the
US in 2010 (California Department of Food and Agriculture 2011) and an exponential regression using 2001–2013
CFDA data estimates the number of dairy cows in California may increase by a factor of 1.5 by the year 2050. Methane
emission rates were estimated from GHG inventory Documentation (California Air Resouces Board 2014) for each
manure management practice: liquid/slurry, anaerobic lagoon, anaerobic digester, daily spread, deep pit, pasture, and
solid storage. The increase in the cow population was assumed to occur uniformly across all management practices
except for the systems used in biogas capture. These systems, including anaerobic digester, anaerobic lagoon, and
liquid/slurry management practices, were adjusted to meet the quantities of biogas specified by each CA-TIMES
scenario. The amount of waste produced by each dairy cow each year was used to estimate the annual biomethane
production and energy potential of each animal. The electricity potential from biomethane is then calculated using
AgSTAR conversion rates (Environmental Protection Agency 2010, U.S. Environmental Protection Agency AgSTAR
Program 2011). The overall fugitive VOC emissions from animal waste declines in the biogas production scenarios
since a large fraction of the waste is treated. Overall, fugitive dairy manure VOC emissions increased by 50% due to
cow population growth in the BAU scenario, and decreased by a factor of a 33% for the GHG-Step scenario relative
to 2010.
Future biomethane production sites were selected based on recommendations from the USDA's Cooperative
Approaches for Implementation of Dairy Manure Digesters (U.S. Department of Agriculture Rural Development
Agency 2009). Mainly, locations were selected with nearby pipeline networks (Gilbreath, Rose et al. 2014) to
transport raw biogas to a centralized clean-up facility, where it can then be compressed and sold for use by electric
generation power plants or transportation fuels. This was considered a more viable option as natural gas pipeline
infrastructure is easy to access, demand from electric utilities for biomethane is high to meet the renewable portfolio
standard (RPS), and a centralized clean-up facility is more economical than distributed facilities.
**3 Results and Discussion**
**3.1 On-Road Mobile Emissions**
Figure 10 illustrates particulate matter emissions of tire and brake wear from on-road vehicles under the BAU and
GHG-Step scenarios. The fine spatial distribution of the emissions reflects the spatial distribution of tire and brake
wear emissions in the base 2010 inventory that is updated using EMFAC predictions to produce the intermediate
2050 emissions inventory. The technology changes inherent in the CA-TIMES BAU and GHG-Step scenarios are
then applied uniformly across the state yielding virtually identical spatial distributions for the final 2050 BAU and
GHG-Step scenario emissions. Tire and brake wear emissions patterns illustrated in Figure 10 essentially follow
predicted vehicle activity patterns in the state. Predicted emissions are highest in major urban centers and along
major transportation corridors. Although increase in vehicular activity was part of this study, expansion of
roadways between 2010 and 2050 were not considered in this study and may be updated in newer versions of the
model.
California's environmental regulations apply uniformly across the state, which supports the assumption of uniform
GHG emissions reductions for on-road vehicles. Despite the uniform regulatory landscape, some of the measures
described in the CA-TIMES GHG-Step scenario rely on modified behavioral patterns and willingness or ability to
adopt new technologies, which may change by region. Education levels, personal wealth, and environmental
attitudes vary sharply across California. Capturing these trends in sub-regions of the state will require surveys of
consumer choice and predictions of future behavior that are beyond the scope of the current manuscript.

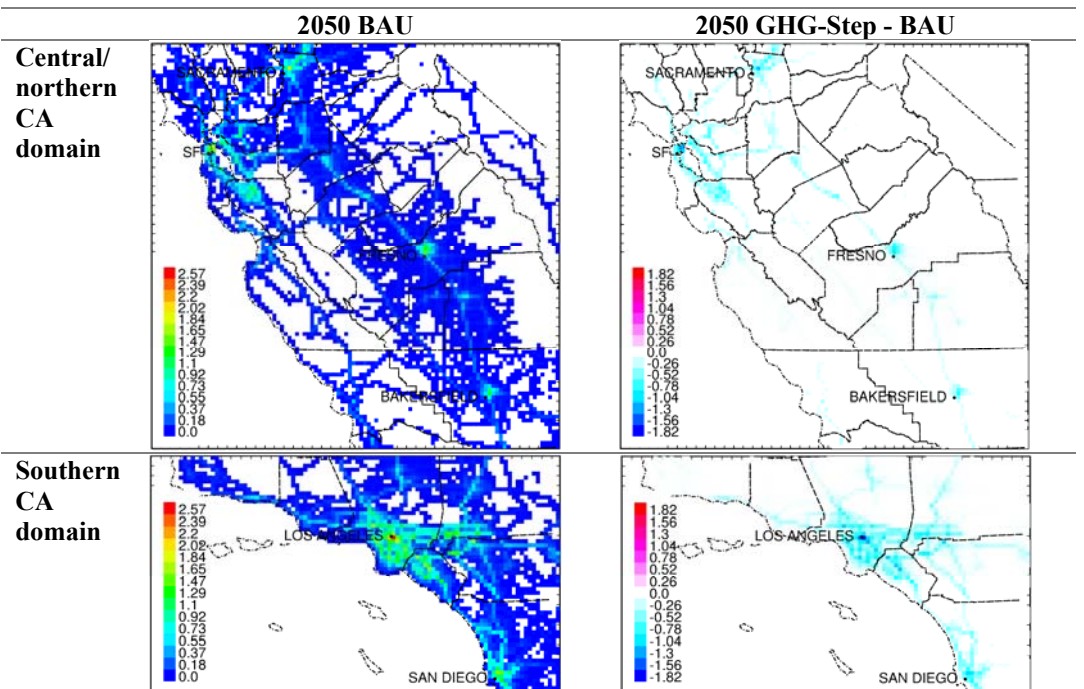

**Figure 10: Particulate matter emissions from vehicle tire and break wear in the BAU scenario (left panels) and emissions**
**change in the GHG-Step scenario (right panels). Units are μg m$^{-2}$ min$^{-1}$.**
Figure 11 illustrates the particulate matter emissions from tailpipe exhaust under the 2050 BAU scenario and the
2050 GHG-Step scenario. Similar to the tire and brake wear emissions, the spatial pattern for mobile sources is
identical under both scenarios because the technology changes specified by the CA-TIMES model are applied
uniformly over the entire state. Tailpipe particulate matter emissions once again follow patterns of vehicle activity
as predicted by EMFAC. Of greater interest is the prediction that tire and brake wear emissions (Fig. 10) will
exceed tailpipe emissions (Fig. 11) in both the 2050 BAU and GHG-Step scenarios due to the adoption of
increasingly clean vehicle technology. Tailpipe emissions in the GHG-Step scenario are a factor of ~1.8 lower than
tailpipe emissions in the BAU scenario. In contrast, tire and brake wear emissions are predicted to decrease by a
factor of +3 under the GHG-Step scenario.  This reflects the fact that BAU gasoline and diesel tailpipe emissions
already incorporate significant emissions control technology yielding fewer opportunities for further improvement.
Tire and brake wear emissions have almost no control technology in the BAU scenario, which makes the widespread
adoption of electric or hybrid drivetrains using regenerative braking particularly effective at reducing emissions.
The current analysis assumes that no new major highways will be built in California and population growth is
accommodated partially through increased urban density such that traffic volumes increase uniformly across the
transportation network.  These assumptions are simplistic but a previous study of smartgrowth in the San Joaquin
Valley indicated that more detailed accounting of population growth had minimal impact on air quality (Hixson,
Mahmud et al. 2010).

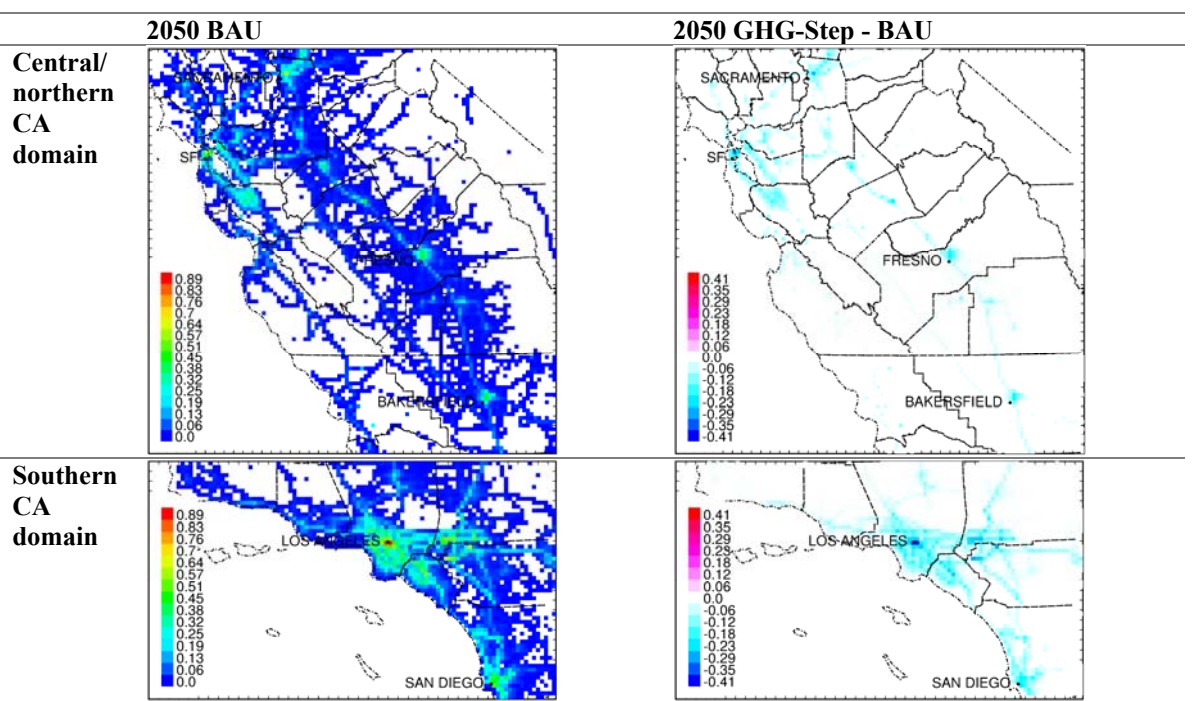

**Figure 11: Particulate matter emissions of vehicle tailpipe exhaust in the BAU scenario (left panels) and emissions change**
**in the GHG-Step scenario (right panels).  Units are μg m$^{-2}$ min$^{-1}$.**

**3.2 Rail, and Off-Road Emissions**
Particulate matter emissions from off-road and rail sources are plotted in Fig. 12 for the BAU and GHG-Step
scenarios examined in the current study.  Maximum statewide particulate matter emissions for this source category
are centered at the location of major construction projects with lower emissions rates for "routine" off-road
emissions distributed more broadly according to typical activity patterns for smaller construction projects, rail, etc.
The 2010 emissions inventory that acts as the basis for the 2050 projections in the current project correctly identified
replacement of the east span of the Bay Bridge in the San Francisco Bay Area as the leading construction project
with the highest overall emissions in the state.  This ~$6.5B project spanned more than 10 years with the new bridge
completed in 2013 and final decommissioning and demolition of the old eastern span scheduled for 2018.
It is difficult to predict the location of major construction projects in 2050 but it is reasonable to expect that several
large projects will be active in that timeframe.  Candidate projects currently under discussion include additional
replacement of California's numerous highways and bridges, upgrading California's water conveyance systems to
better withstand earthquakes, development of high speed rail lines, reinforcement or expansion of seawalls to protect
property, etc.  Each of these projects will potentially emit criteria pollutants that would affect air quality over major
urban centers.  In the present study, the peak emissions associated with the major construction project around the
Bay Bridge were retained in the future scenario as an example of a major construction project near an urban area.
Future model analysis that uses these emissions should conduct sensitivity tests to ensure that the assumed
placement of this example major construction project does not influence the overall conclusions of the study.
Maximum particulate matter emissions shown in Fig. 12 decrease by a factor of approximately 1.6 4in the GHG-
Step scenario relative to the BAU scenario.  Adoption of biomass based fuels was also found to reduce emissions of
$SO_x$, HC, PM, and occasionally CO from off-road and rail sources, but $NO_x$ emissions increased for some fuel
choices.

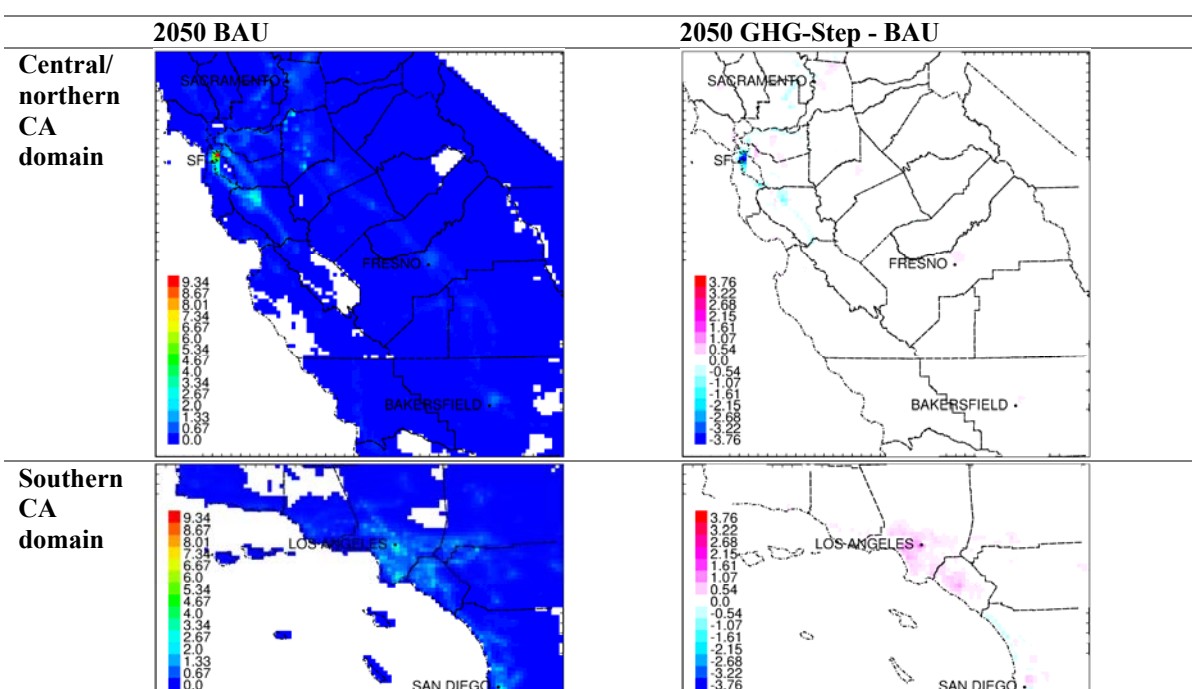

**Figure 12: Particulate matter emissions from rail and other off-road sources in the BAU scenario (left panels) and**
**emissions change in the GHG-Step scenario (right panels).  Units are µg m$^{-2}$ min$^{-1}$.**

**3.3 Marine and Aviation Emissions**
Particulate matter emissions from marine and aviation sources are shown in Fig. 13 for the BAU and GHG-Step
scenarios considered in the present study.  The highest particulate matter emissions rates occur in off-shore shipping
lanes that converge on the Port of Los Angeles, the Port of Long Beach, and the Port of Oakland.  Emissions rates
change with proximity to California shores due to regulations governing sulfur content of marine fuel or ship speed.
Emissions patterns at inland locations reflect shipping activity on inland waterways or activity surrounding small
regional airports.

Maximum particulate matter emissions rates from marine sources increase under the GHG-Step scenario as illustrated
most clearly in the right panels of Fig 13.  CA-TIMES determined that the available biofuel capacity could be more
efficiently used to offset traditional fossil fuels for on-road transportation sources and so the GHG-Step scenario is
predicted to incorporate additional fossil fuels for marine sources under the GHG-Step scenario vs. the BAU scenario.
The net result of the disbenefits associated with increased marine emissions vs. the benefits of the decreased on-road
emissions will be considered in future studies that include analysis with regional air quality models.

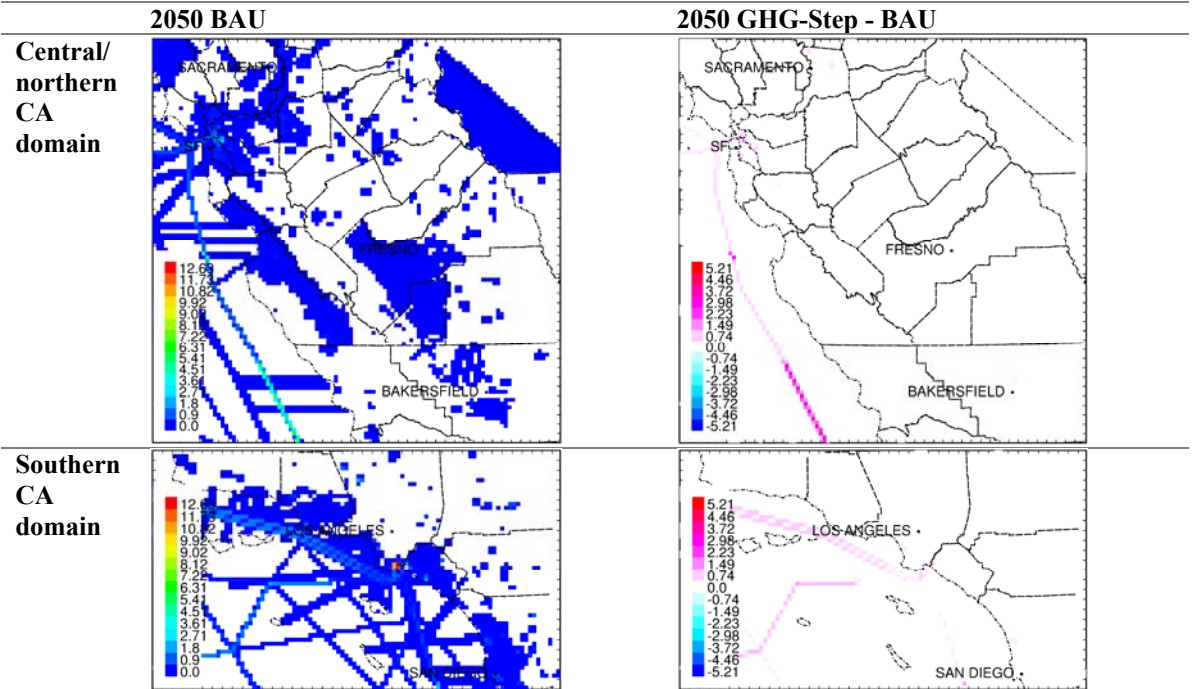

**Figure 13: Particulate matter emissions from marine and aviation sources in the BAU scenario (left panels) and emissions**
**change in the GHG-Step scenario (right panels).  Units are μg m$^{-2}$ min$^{-1}$.**

**3.3 Residential and Commercial Emissions**
Fig. 14 illustrates particulate matter emissions from residential and commercial sources under the 2050 BAU and
GHG-Step scenarios.  The spatial patterns of emissions largely follow the estimated population projections in
California in the year 2050 as summarized in Table S24.  Population growth was assumed to be identical under the
BAU and GHG-Step scenarios yielding virtually identical spatial distributions for both scenarios.  The adoption of
new technologies and altered behavioral patterns predicted by the CA-TIMES model under the GHG-Step scenario
were applied uniformly over the state without modification by income, education level, or regional differences in
environmental attitudes.  Predicted changes to particulate matter emissions from residential and commercial sources
are modest with slight reductions of ~10% mostly attributed to energy efficiency measures.  Widespread adoption of
biomethane to replace natural gas is predicted in the GHG-Step scenario but this fuel change has little impact on
criteria pollutant emissions.

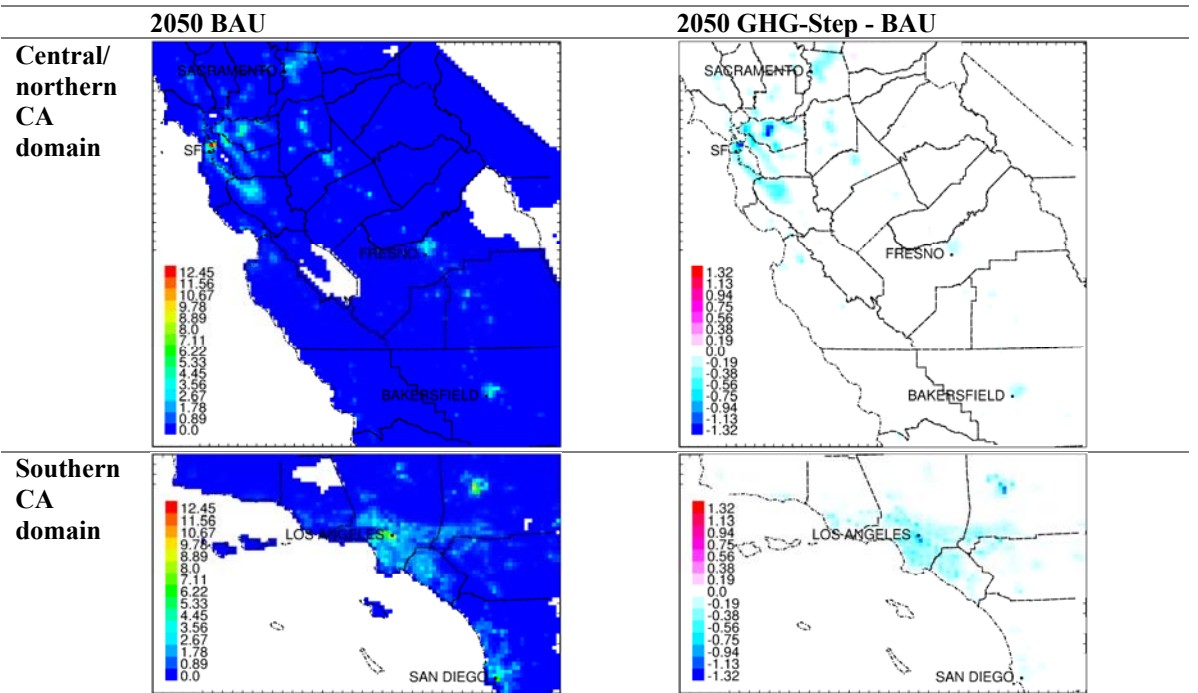

**Figure 14: Particulate matter emissions from residential and commercial sources in the BAU scenario (left panels) and**
**emissions change in the GHG-Step scenario (right panels).  Units are μg m$^{-2}$ min$^{-1}$.**

**3.4 Electricity Generation Emissions**
Fig. 15 illustrates predicted emissions of particulate matter from combustion processes used to generate electricity.
These emissions are represented as point sources and so only the grid cell containing an electrical generation unit are
colored.  The highest emissions rates for individual grid cells are associated with a small number major electrical
generation stations typically powered by natural gas in the BAU scenario.  The majority of the colored grid cells in
Fig. 15 are associated with smaller backup generators that operate intermittently and therefore have very low
emissions.  These backup units are typically powered by a fossil fuel such as diesel fuel in the BAU scenario, with a
shift to biofuels in the GHG-Step scenario.  This fuel switch has modest impact on total emissions given the low
utilization of these units.
Peak emissions rates of particulate matter in the GHG scenario decrease by a factor of ~1.7 in the GHG-Step
scenario primarily due to a reduction in fossil fuel electricity generation in favor of a shift to solar and wind sources
(see Fig. 9). All generating stations are assumed to continue operation at a reduced rate in the GHG-Step scenario
rather than selectively decommissioning some stations. The age and efficiency of existing natural gas generating
stations will likely be key factors determining how they are operated in the future scenarios. Solar and wind
electricity generation does not emit criteria pollutants and so the location of these facilities is not shown in Fig 15.

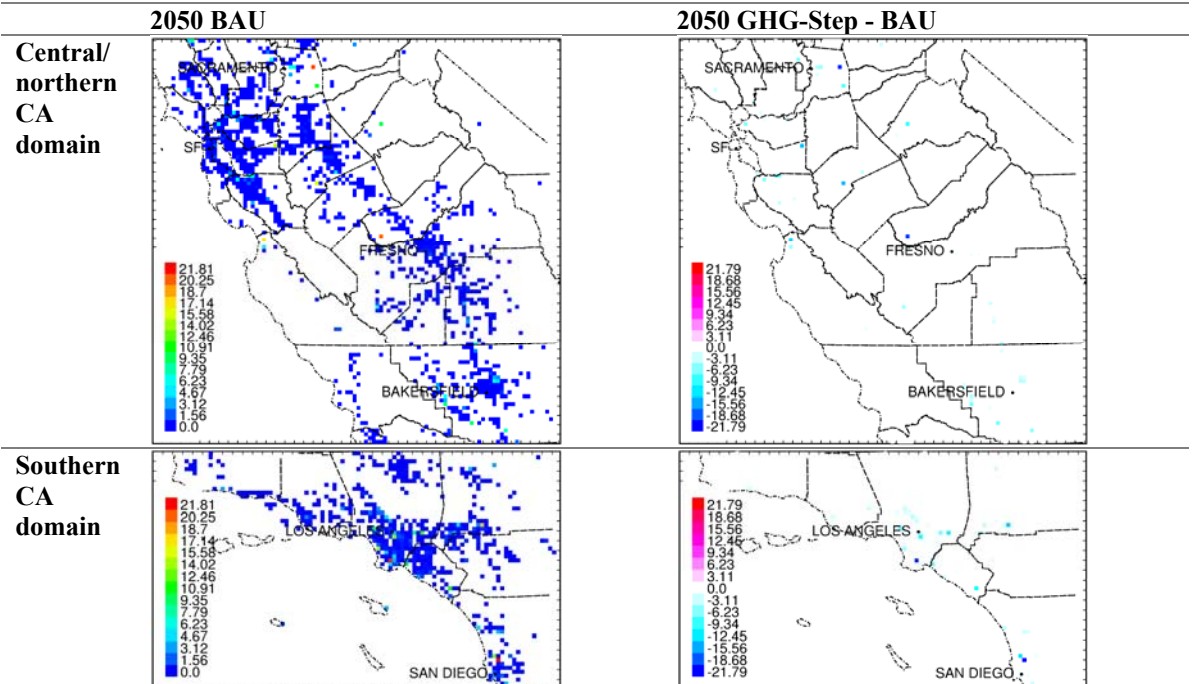

**Figure 15: Particulate matter emissions from electricity generation (emission source category type 6) in the BAU scenario**
**(left panels) and emissions change in the GHG-Step scenario (right panels). Units are μg m$^{-2}$ min$^{-1}$.**

**3.5 Biorefinery Emissions**
Figure 16 shows the locations of refineries producing biofuels (bio-refineries) in California under the BAU and
GHJG-Step scenarios considered in the present study. The location of future bio-refineries was chosen to minimize
transportation costs for the raw materials feeding into the refinery and the delivery of fuel to the final point of end-
use. Additional zoning constraints were considered to prevent the placement of bio-refineries near schools, hospitals
or other locations with sensitivity populations. More generally, a constraint was considered to restrict the placement
of new bio-refineries in regions that currently violate the NAAQS. The top panels of Fig. 10 therefore do not allow
the placement of bio-refineries in either the SJV or the SoCAB, while the less constrained scenarios illustrated in the
lower panels of Fig. 16 do not impose this restriction. In practice, bio-refineries were generally sited near landfills,
industrial, or agricultural areas within each city selected as economically optimal within the specified constraints.
The enforcement of NAAQS constrains on bio-refineries lead to a smaller number of larger refineries under both the
BAU and GHG-Step scenarios.  Note that overall bio-refining output is higher in the BAU scenario than in the
GHG-Step scenario.  Bio-fuels have lower associated GHG emissions than traditional fossil fuels but their carbon
intensity is still too high to meet the GHG emissions target represented in the GHG-Step scenario.  The CA-TIMES
model therefore predicts that a portion of the energy supplied by biofuels in the BAU scenario will be supplied
instead by wind and solar in the GHG-Step scenario.

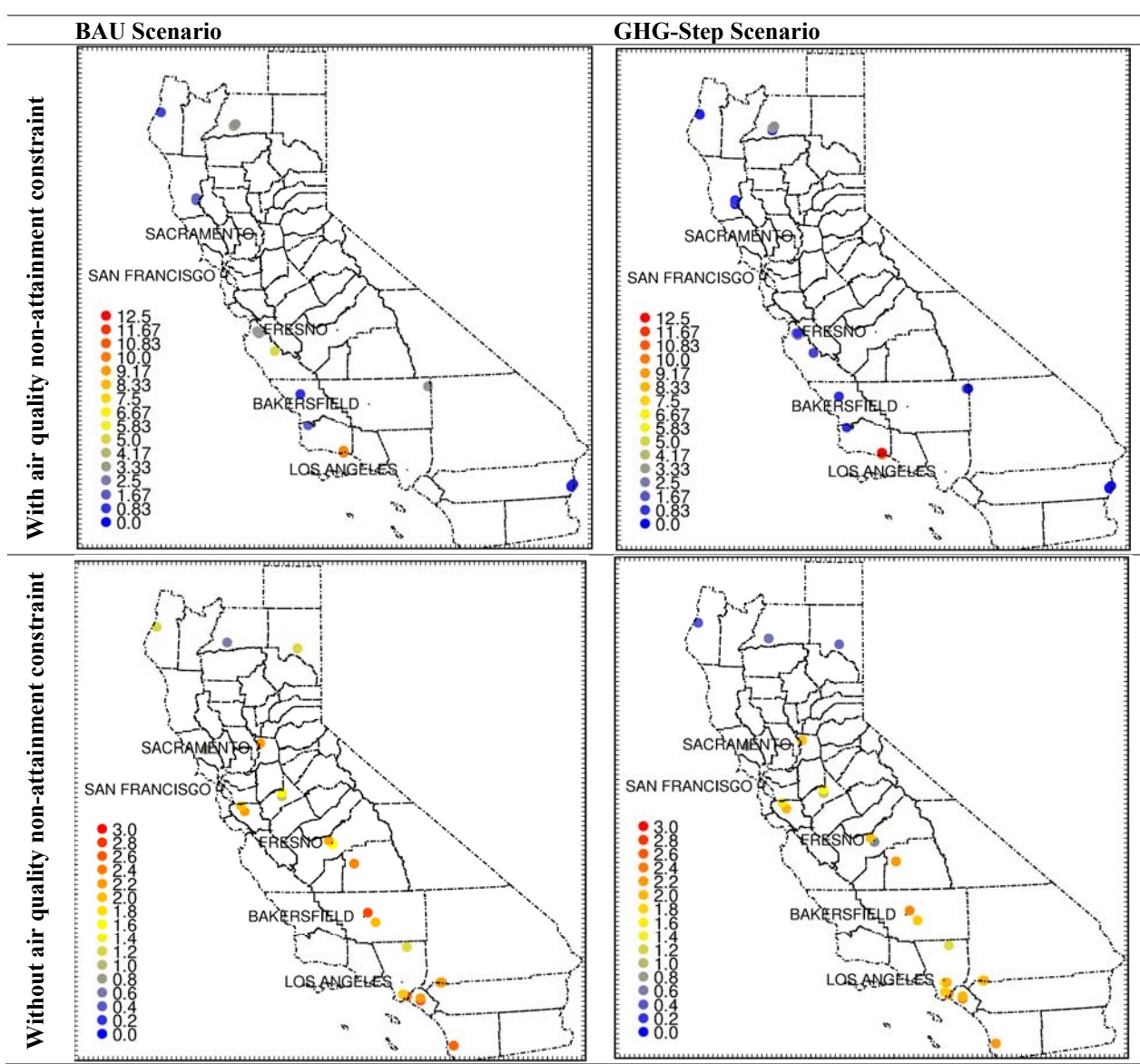

**Figure 16: Biorefinery locations under the BAU scenario (left column) and the GHG-Step scenario (right column).**
**Legend shows PM2.5 mass emission rates per facility in μg m$^{-2}$ min$^{-1}$. Top panels represent the constrained case where**
**biorefineries cannot be located in air basins out of compliance with National Ambient Air Quality Standards (NAAQS).**
**Bottom panels are not constrained by NAAQS status.**

**3.6 Summary of Statewide Emissions**
Fig. 17a illustrates the net change in emissions related to criteria pollutants in California in the GHG-Step scenario
vs. the BAU scenario analyzed in the current study. Emissions of each pollutant are broken down by the major
emissions categories analyzed in Section 2. The miscellaneous category is equivalent in the BAU and GHG-Step
scenarios and hence is not plotted. Contributions below 0% indicate emissions reductions, while contributions
above 0% indicate emissions increases. Each of these changes represents the statewide average for the sources
within the indicated sector. Note that the changes within each sector may not be uniform across the entire state. The
net change in total emissions is indicated by the black horizontal line for each species. It is immediately apparent
that the emissions reductions illustrated in Fig. 17a are not uniform for all pollutants. Maximum reductions of ~60%
are observed for $CO_2$ and particulate copper (Cu) emissions. In contrast, emissions of particulate $SO_4^{2-}$, gaseous CO
and gaseous SOx actually increase under the GHG-Step scenario due to tradeoffs in the technologies adopted in the
off-road mobile categories (rail, marine, aviation, etc) needed to optimize the overall GHG emissions across the
state. Emissions of pollutants that experience increasing trends in Fig. 17a are minor in the present-day inventory
and so that they do not currently trigger NAAQS violations. Changes in key, highly emitted pollutants fall in
between the extreme cases described above (see results for particulate EC, particulate OC, and gaseous NOx). Each
of these pollutants experience a net decrease in total emissions averaged across California, but emissions changes
are not uniform across all categories. Some technology and fuel changes cause higher emissions which are offset by
savings in other categories. This complex mixture of tradeoffs reflects the optimal economic approach to GHG
reductions determined by the CA-TIMES model.
The changing activity patterns, fuels, and technologies included in the GHG-Step scenario lead to changes in the
emitted particle size and composition distribution. This leads to differences in the response of primary particulate
matter with aerodynamic diameter less than 2.5 μm (PM2.5) and less than 0.1 μm (PM0.1; ultrafine particles).
Ultrafine particles are an emerging pollutant of concern expected to influence public health (Delfino, Sioutas et al.
2005, Knol, de Hartog et al. 2009, Hoek, Boogaard et al. 2010). The results shown in Fig. 17a illustrate that the
GHG-Step scenario leads to only a 4% decrease in primary PM2.5 emissions but a much larger 36% reduction in
PM0.1 emissions. Recent epidemiology results indicate that PM0.1 is associated with mortality in the California
Teachers Study (Ostro, Hu et al. 2015). Enhanced PM0.1 emissions reductions could amplify the potential health
benefits of the future GHG-Step scenario beyond the level expected from PM2.5 emissions reductions.
Fig 17b. shows the net change in criteria pollutant emissions predicted using the expert analysis approach described
by Shindell et al. (2012). These results are presented as a comparison point to the results illustrated in Fig. 17a and
listed in SI Table S36 through Table S38. The expert analysis scenario focused on a small number of measures
targeted for countries which are in the early stages of adopting policies to reduce GHG emissions or mitigate
regional air quality problems. As a result, the measures described by Shindell et al. have a large impact on global
public health but they will have a very minor impact on California (or any other major state or country that has
already implemented significant emissions controls).
Comparison of Fig. 17a and Fig. 17b illustrates that only reductions in particulate EC are comparable in the Shindell
et al. and CA-TIMES scenarios due to the mitigation of emissions from off-road diesel engines. CA-TIMES
accomplishes this reduction through a combined switch in fuels and adoption of diesel particle filters on remaining
diesel and bio-diesel sources to achieve a combined reduction in GHG emissions and criteria pollutant emissions.
Shindell et al. assume uniform adoption of diesel particle filters on all off-road diesel engines with no fuel
switching. Shindell et al. also specify the adoption of digesters for dairy waste and increased use of landfill gas as
renewable methane sources. CA-TIMES predicts similar adoption resulting in a ~35-40% reduction in ammonia
($NH_3$) emissions from these sources. The CA-TIMES approach considered in the present study additionally
considers how the emissions of bio-methane differ from the emissions of traditional natural gas. The only other
significant measure specified by Shindell et al. that could reduce criteria pollutant emissions in California is a
complete ban on burning of agricultural waste. California already limits agriculture burns to avoid stagnation
periods. Thus, even the apparent savings associated with reduced agricultural burns apparent in Fig. 17b are likely
to have limited practical impact on air quality in the state. Shindell et al. do not consider the adoption of low carbon
fuels or electrification of on-road vehicles which are necessary to achieve deep GHG reductions in CA.
Overall, the analysis presented by Shindell et al. (2012) is appropriately targeted at global health but the measures
considered in this analysis do not achieve California's GHG objectives and the criteria pollutant emissions changes
associated with them will not support calculations for future air quality in California. Energy economic models such
as CA-TIMES represent a more realistic tool for development of scenarios in regions like California that have
already considered all simple measures. Careful analysis is required to understand the resulting complex pattern of
tradeoffs between emissions in different categories that results from these scenarios.

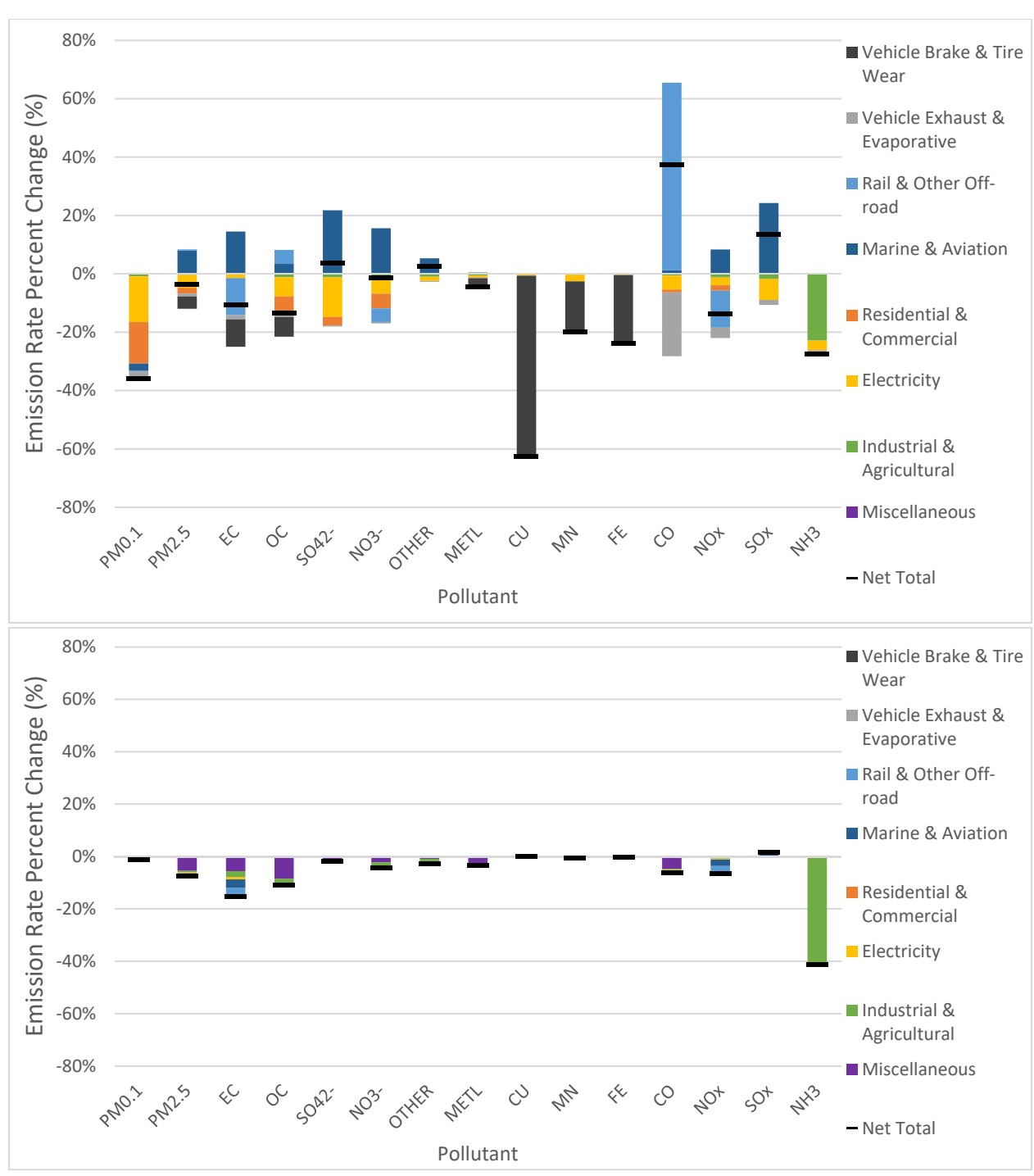



**Figure 17: Change in pollutant emission rate relative to BAU scenario. Panel (a) represents GHG-Step analyzed in the current study using the CA-TIMES model. Panel (b) represents expert analysis presented by Shindell et al. (2012).**

Fig. 18 illustrates examples of spatial patterns of emissions changes under the GHG-Step scenario predicted by CA-TIMES in the current study. The offsetting increasing and decreasing emissions changes illustrated in Fig. 17 do not occur uniformly over the state but instead appear as regions of localized increasing and decreasing emissions. As an even greater complication, the spatial pattern of increasing and decreasing emissions changes for each pollutant.

The top panels of Fig. 18 illustrate changes in the commercial and residential sector for NOx emissions (Fig 18a)

and OC emissions (Fig 18b) in central California. Patterns of emissions increases or decreases are similar in major
urban centers (San Francisco and Sacramento) but different patterns are predicted for emissions of NOx and OC in
the heavily polluted San Joaquin Valley (Fresno and Bakersfield). The lower panels of Fig. 18 illustrate even
stronger variation in the spatial pattern of emissions changes in the off-road and rail categories in southern
California. The spatial pattern of the change in particulate EC emissions (Fig. 18c) differs strongly from the spatial
pattern of the change in particulate OC emissions (Fig. 18d).
All of the emissions illustrated in Fig. 18 will produce regions of increased or decreased pollutant concentrations.
Given that each region is highly populated, these emissions patterns will have a direct effect on population exposure.
Detailed analysis with regional air quality models at a resolution of 4km or finer will be required to understand the
health implications of these changing emissions. California requires this level of fine-scale emissions analysis to
accurately predict the air quality impacts of future GHG mitigation strategies in the state. Similar efforts will be
required to analyze the effects of GHG mitigation strategies on criteria pollutants in other highly-populated regions
that have already moved beyond simple emissions regulations banning obvious sources of air pollution.

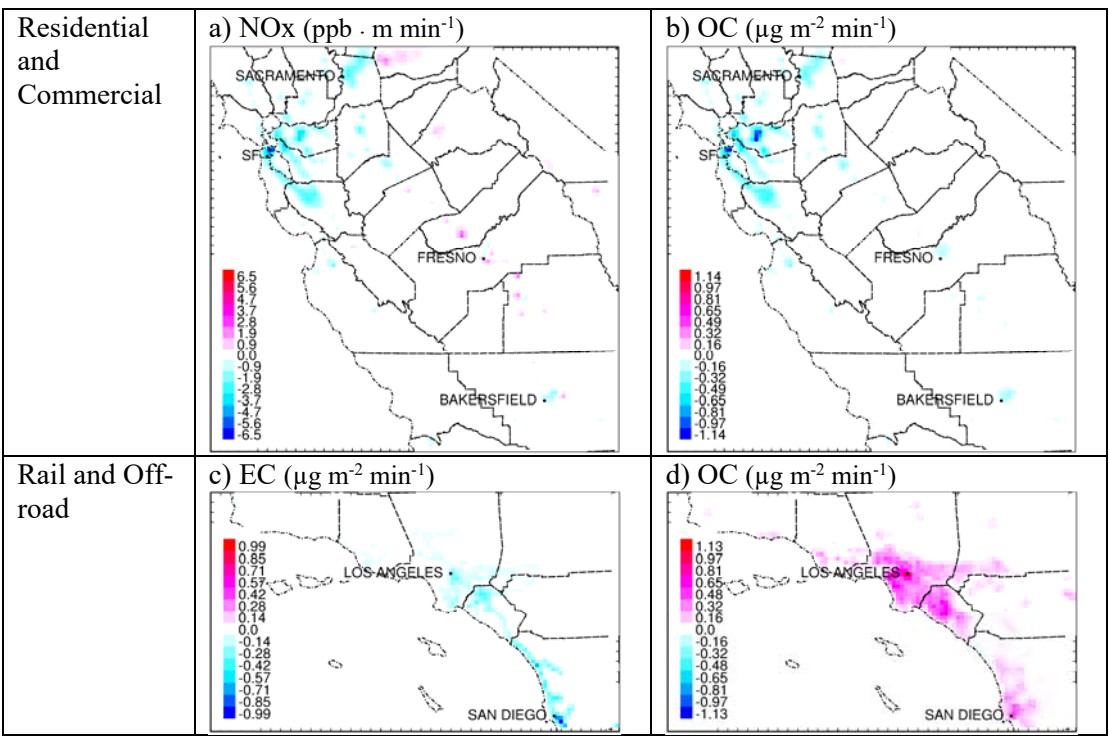

**Figure 18: Change in emissions in the GHG-Step scenario relative to the BAU scenario . (a) NOx from**
**residential and commercial sources (ppb · m min$^{-1}$), (b) particulate OC from residential and commercial**
**sources (µg m$^{-2}$ min$^{-1}$), (c) particulate EC from off road and rail sources (µg m$^{-2}$ min$^{-1}$), and (d) particulate OC**
**from off road and rail sources (µg m$^{-2}$ min$^{-1}$).**
The CA-REMARQUE projections for criteria pollutant emissions associated with optimal climate policies in
California should not be directly extrapolated to other regions or countries. Instead, the methods used by CA-
REMARQUE should be applied to each new region to fully consider the appropriate energy resources available,
consumption patterns, equipment vintages, aftertreatment regulations and population and economic growth rates.
Each region may have a different optimal set of GHG mitigation technologies and policies that will lead to different
rates and spatial patterns of emission compared to the changes predicted in California.  Many developing regions
will be able to select less expensive GHG mitigation strategies that also reduce GHG and criteria pollutant emission
relative to their BAU scenario.  Within developed regions such as other U.S. states, the elements of the mobile
emissions inventory maintained by the U.S. EPA (MOVES and mobile portion of the National Emissions Inventory)
can be adapted to replace the corresponding California information (EMFAC, mobile portion of the CARB
inventory).  Changes to off-road emissions would need to be estimated following procedures similar to those
employed in the CARB off-road VISION model.  Effort would be needed to estimate how changes to marine fuel
sources would influence emissions at major ports.  Studies would need to be conducted describing potential
locations for new facilities producing low-carbon fuels and the resulting emissions from those facilities.  This
information would support a fully resolved analysis of the criteria pollutant emissions associated with climate
policies outside of California.

## 4 Conclusions

The California REgional Multisector AiR QUality Emissions (CA-REMARQUE) model has been developed to
translate optimized GHG mitigation policies to criteria pollutant emissions in California.  Minimum-cost GHG
policies are first selected with the energy economic model CA-TIMES.  Tailored methods are then used to predict
corresponding changes in criteria-pollutant emissions for individual categories including on-road vehicles, off-road
vehicles, marine, aviation, rail, residential, commercial, electricity generation, industrial, and agricultural emissions.
Translation methods account for efficiency improvements, changing technology, and changing fuels with
corresponding changes to criteria pollutant emissions.  Modifications to the composition of reactive organic gases
and the size and composition of airborne particulate matter are considered.  Translation methods also account for
increased emissions associated with some measures, such as the need to produce new bio-fuels including bio-diesel,
ethanol, and hydrogen.
The CA-REMARQUE model is demonstrated by predicting emissions in 2050 under a Business as Usual scenario
(BAU) and an optimized GHG mitigation scenario (GHG-Step) in California.  The results show that the optimal
scenario for GHG mitigation produces increasing criteria pollutant emissions in some categories that are offset by
decreases in other categories.  These tradeoffs yield a complex pattern of emissions trends with sub-regions of
increasing emissions and sub-regions of decreasing criteria pollutant emissions across California when viewed at
4km spatial resolution.  In contrast, a simplified expert analysis scenario designed to address global GHG emissions
may not necessarily reduce criteria pollutant emissions in California because many emission sources have already
been controlled by the state's air pollution regulations.  The expert analysis method does not consider complex fuel
switching scenarios beyond the replacement of natural gas with biomethane. Choosing an economically optimal
scenario of additional measures needed to achieve GHG mitigation goals in California requires tools beyond expert
analysis opinions.  Likewise, fully accounting for the corresponding changes to criteria pollutant emissions requires
sophisticated analysis in fully developed countries and states with strict existing environmental regulations.
The California sub-regions of increasing and decreasing criteria pollutant emissions predicted in the current project
occur in close proximity to major population centers and so they will almost certainly influence population exposure
and public health.  The emissions inventories created in the current study will be analyzed using regional air quality
models in a future study to fully calculate impacts on public health.
**4 Code and Data Availability:**
All of the data necessary to calculate changes to emissions inventories are published in full in the main text and
supporting information section of the manuscript.  Collaborators may request the CA-REMARQUE model code or
final criteria pollutant emissions inventories by contacting the corresponding author.  Note that the CA-
REMARQUE v1.0 model is separate from the CA-TIMES energy-economic model.
**5 Acknowledgments:**
This study was funded by a National Center for Sustainable Transportation Dissertation Grant and the United States
Environmental Protection Agency under Grant No. R83587901. Although the research described in the article has
been funded by the United States Environmental Protection Agency it has not been subject to the Agency's required
peer and policy review and therefore does not necessarily reflect the reviews of the agency and no official endorsement
should be inferred.

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
