# Peer review of "Estimating Criteria Pollutant Emissions Using the California"

_Geoscientific Model Development, 2017_

## Referee Comment (RC1) · Anonymous Referee #1 · 7 Sep 2017

———— General comments: ————

In this manuscript, the authors use the CA-REMARQUE method to develop 4km gridded emission inventories for the year 2050. Two inventories are generated: a baseline inventory and one in which GHG emissions are reduced by 80% relative to 1990 levels. The scenarios were developed using the CA-TIMES model. The authors describe their approach for translating the CA-TIMES projections into criteria pollutant values based on a variety of sector-specific procedures. The authors then examine the resulting 4 km inventories and highlight differences in scenario-, sector-, and pollutant-specific

emission trends. Furthermore, they compare the changes in emissions to those of a more generalized, expert-driven approach and suggest that considering California-specific conditions (e.g., regulatory environment, existing stock, renewable resources) yield very different changes in emissions than a more generalized approach found in the literature.

Scientific signficance: This work tackles an important objective - developing emissions inventories that can be used to examine the air quality implications of specific energy system scenarios. While others have also tackled this problem, this work adds important detail. However, because the literature review is very limited, the authors are not able to explicitly identify how they advance the science. (see "Specific Comments" for suggestions about expanding the literature review)

Scientific quality: I believe that the scientific approach and methods are underlying the work presented here are valid.

Scientific reproducibility: The authors do an very good job of describing the process that they used to develop future-year inventories from energy system modeling results. While replicating the work for this or another state or geographic region would undoubtedly be a large and difficult task, that difficulty would not be due to lack of information on the method.

Presentation quality: In general, I feel as if this manuscript is well written and that the experimental design and results reinforce the arguments presented by the authors. Nonetheless, I think the presentation quality could be improved substantially if the graphics provided in the results section were revised. One particular area of improvement is in the graphics that map PM emissions for both scenarios. I believe the intent of these graphics was to illustrate (i) the ability to develop spatially explicit inventories, and (ii) the changes in these emissionsfrom one scenario to another. By using different scales on each image, however, the differences are not readily apparent. I suggeste using the same scale or perhaps showing a graphic for the business as usual

case and another with deltas associated with the GHG mitigation case. The stacked bar graphs that showed pollutant specific changes were also confusing and it was not clear what stacking of percents was intended to indicate. Those data could be much more easily presented and compared using a table.

———— Specific comments: ————

I feel that the manuscript has several deficiciencies, and that it could be improved substantially if these deficiencies are addressed. Please see the "Specific Comments" below. My main concern is related to the literature review which could be expanded. I suggest that the authors include addition studies that examine the emissions or air quality impacts of alternative policy scenarios. A few for the U.S. are listed below (although I do not think that all of these necessarily need to be referenced):

* Keshavarzmohammadian A, DK Henze, and JB Milford (2017). Emission impacts of electric vehicles in the US transportation sector following optimistic cost and efficiency projections. Envir. Sci. Technol., 51(12), 6665-6673.

* Loughlin DH, WG Benjey, CG Nolte (2011). ESP v1.0: methodology for exploring emission impacts of future scenarios in the United States. Geosci. Model Dev., 4, 287-297.

* Ran L, DH Loughlin, D Yang, Z Adelman, BH Baek, and CG Nolte (2015). ESP v2.0: enhanced method for exploring emission impacts of future scenarios in the United States - addressing spatial allocation. Geosci. Model Dev., 8, 1775-1787.

* Rudokas J, PJ Miller, MA Trail, and AG Armistead (2015). Regional air quality management aspects of climate change: Impact of climate mitigation options on regional air emissions. Environ. Sci. Technol., 49(8), 5170-5177.

* Trail MA, AP Tsimpidi, P Liu, K Tsigaridis, Y Hu, JR Rudokas, PJ Miller, A Nenes, and AG Russell (2015), Impacts of potential CO2-reduction policies on air quality in the United States. 49(8), 5133-5141.

Similar to this manuscript, the Loughlin et al. 2011 paper also illustrates the development of region-, sectoral- and pollutant-specific implications of a climate policy and discusses how an energy system model's emission projections can be used to develop inputs to air quality modeling. Ran et al. (2015) build on that by adding a land use change compontent to spatially re-allocate emissions for some sectors. This manuscript would benefit greatly by a comparison to the Loughlin et al. and Ran et al papers. In such a comparison, I feel that the work presented here has much to add. For example, it explicitly tackles a state's regulations, goes into much greater detail regarding specific sectors (e.g., nonroad and marine), tackles the siting of new biomass-related emission sources, examines speciated emissions changes from fuel switching (e.g., fossil to biofuels), incorporates consideration of the PM benefits of regenerative braking, and seeks to examine the impact of controls on particle size distribution. Described in the context of these advances, I think the merits of the work presented here are much more clear and point to areas where other analyses could be improved.

I have an additional concern that is perhaps less critical. My concern is related to the presentation of the scenarios themselves. The first scenario is relatively easy to understand as it is just a baseline or business as usual scenario. Nonetheless, California has a very unique energy system, so the underlying trends and dynamics there may not be readily apparent to readers. The second scenario is a GHG reduction scenario, although how the GHG constraint is implemented is not completely clear. Since the authors only compare emissions between a base year, 2010, and a future year, 2050, they have chosen to show the CA-TIMES results only for those years. To convey the scenarios and underlying dynamics more fully, however, I believe the authors should show several model outputs for the period from 2010 *through* 2050, not just in 2010 and 2050. These outputs would include at a minimum, and for both scenarios: (i) $CO_2$ emissions by sector, and (ii) electricity production by technology, and (iii) energy system fuel use. Additionally, displaying fuel used or technologies adopted in the transportation and transportation sub-sectors could be of interest but are not necessary. Stacked area plots would be one form of presenting these graphics. The sectoral GHG graphic

would be particularly useful in understanding how the GHG policy is represented in the model. For example, is it a constraint represented only from 2050 onward as the text seems to imply? If so, how did the model respond to such a shock, and did it begin to make structural changes to the energy system with foresight or respond in more of a myopic way? These results wouldn't necessarily need to be in the main body of the manuscript, and instead could be in the supplemental information.

Additional substantive comments are listed below, preceded by page number:

55: The text refers to other modeling efforts as over-simplifying the California economy. This statement, I think, implies that CA-TIMES includes a much more detailed representation of the economy. I advise care with that description, however. I suspect that CA-TIMES respresents the "energy economy" only, which is only a portion of the larger economy, and may represent interactions with the rest of the economy via simple elasticities for energy demands or perhaps with a function that links to regional GDP. Many readers will assume that a model of the economy would take the form of a Computable General Equilibrium model, which would represent things like tradeoffs between labor and capital, employment, household income, etc.

76: I suggest adding additional information to the description of CA-TIMES. For example, listing the sectors included would be helpful, as would listing the pollutants included in the model. Does it just represent $CO_2$, or does it add other GHGs? What about air pollutants? Are demands fixed or elastic? How the model handles the rest of the country and the rest of the world? How trade with region(s) outside of CA is considered and constrained? Answers to some of these questions are alluded to later in the paper (as we see what CA-TIMES has produced and how it is used), but it would be very useful to have these types of questions answered explicitly when the model is introduced.

Fig 8: What is driving the reduction in commercial energy usage in 2050 GHG-Step? Is it elastic demands? More efficient technologies? Or is there commercial (on-site) solar power that isn't being shown?

Fig 10: At first glance these look the same. It isn't until you examine the legend that it is clear that there has been a 2/3 reduction. I suggest having the scale the same on all the graphs. Alternatively, you could show the 2050 emissions and another graph showing deltas (e.g., where they increase or decrease). The latter approach may be more useful in conveying air pollutant emission co-benefits (or dis-benefits).

689 - Stacking percentages are difficult to interpret? Was a single % change then apportioned to components? If not, it may not be appropriate to stack these values. Perhaps you could show the actual quantities? I feel like a table would convey the information better.

———— Editorial comments and corrections: ————

My other comments, which are more editorial in nature, are listed below. I hope that these are helpful in revising what I feel is a very interesting manuscript.

Line #: comment

54: Editorial suggestion - Here and throughout, avoid using "/". Also, where you have "and/or" I suggest just using "or", which is not necessarily mutually exclusive.

86: CAFE should be spelled without the accent mark on the E

87: References to the regulations affecting CA would be helpful

105: The statement is made that "vehicular emissions for the year 2050 were extrapoled", which left me wondering how. I see that you explain this later in the paper, but is there a way to reorganize so the reader doesn't ask this question, or, if they do, the answer if provided sooner? I had a similar concern with the text "vehicular activity and fuel consumption splits were applied..." in line 108.

Figure 2: The abbreviation EIC is used prior to being defined

Figure 2: Minor suggestion: for the "2050 CA-TIMES Scenario on-road Emissions" box, I suggest renaming this "On-road emissions consistent with a 2050 CA-TIMES

Scenario".

128: A meteorological scaling factor is mentioned, but it isn't clear what is assumed about meteorology in the future.

128: It appears that the methodology does not explicitly consider expansion of existing roadways or the addition of new roadways. Similarly, I did not see a mention of the impact of land use change. If not, perhaps they could be mentioned in the discussion section as ways in which the analysis could be expanded?

145: There are two equations listed as (1). I suggest making each its own.

155: Please provide definitions for the vehicle sub-categories

209: Replace "∼" with "approximately"?

223: "Fueled" misspelled?

345: Perhaps a better way to state "CA-TIMES finds that it is too expensive to adopt biomass-based fuels for ships in the GHG-Step scenario in 2050" is that the model identifies other approaches for meeting the GHG target more cost-effectively?

376: awkward

388: Were wood and distillate held constant, or were they allowed to compete in CA-TIMES... but just not allowed to increase?

Fig 9: Too many significant figures are shown in the table. Perhaps limit to 3 for each value shown?

431: CA-GREET1.8b is argued to have the highest accuracy. But it isn't clear what that means or to what it is being compared to evaluate accuracy.

447: Replace "predicted" with "determined"? CA-TIMES and similar models aren't typically referred to as predictive models. Instead, they are used to evaluate how technonology and fuel choices play out under particular scenarios and assumptions.

450: Do you know the amount of reduced usage vs. switching to imports?

453: Where "Assumed to increase" is used, I think the authors are refering to a model result, not an assumption. Perhaps reword to clarify.

711: The legend for Fig 18a should include at least one more significant digit to be more comparable with the other graphics.

744: You should probably be clear that the CA-TIMES model "code" isn't part of this package (assuming that it is not).

734: "does not have significant impact on criteria..." I understand the point, but this is worded awkwardly.

---

## Short Comment (SC1) · 12 Sep 2017

I would like to point the authors to the program code and data access policy of the GMD journal as outlined in https://www.geoscientific-model-development.net/about/manuscript_types.html . Main objective of the policy is to guarantee reproducibility of the model result presented in paper. In accordance of this objective I would like to encourage to consider uploading your program code and data as supplementary material to ensure persistent access to the data. It would be also useful and interesting for the user to learn about the program environment the model

code is running in and under what license the software can be used.

Lutz Gross GMD Executive Editor

---

## Referee Comment (RC2) · Anonymous Referee #2 · 19 Sep 2017

The authors present an ambitious modeling effort to project air pollutant emissions from multiple sectors under differing climate policies at high resolution across California. In doing so, this study carries out a significant effort to simulate changes to technology, fuels and human activity for varying sources based on multiple models, datasets and careful assumptions. The paper describes a methodology that goes beyond prior attempts to project pollutant emissions changes associated with GHG mitigation strategies. The methodology and model developed will be a valuable contribution to the air quality and climate modeling communities. Additionally, the manuscript is well written.

[Figure]

[Figure]

I believe the paper should be considered for publication in GMD. However, I have some concerns with the manuscript in its current form. These issues must be addressed before I can recommend publication. My comments are described below:

- My largest concern relates to the presentation of results under section 3 Results and Discussion. Although the authors describe the model's results and discuss some interesting findings, the section (particularly the plots and visualizations) should be improved to better communicate the study's results. Figures 10-15 are not informative. Figures 16 and 18 are unclear. The quality of all figures with maps in the section could be improved. Some specific comments are provided below. I encourage the authors to improve the manuscript's overall discussion of results.

-Although the study focuses on California, and the depth with which the authors model projected emissions within the state is a strength, the manuscript would benefit from including a discussion of the potential benefits and challenges associated with applying the emissions modeling methods used to other regions beyond California or at a national scale. Given the specificity of the analysis, it is difficult to identify which elements may be extended to other locations or to Policy/GHG/Energy projections developed with other models beyond CA-TIMES.

- Line 17: Change "play" to "plays"

- Cite references to strengthen the statements in lines 38-40, 40-41 and 42-44.

- The paragraphs from line 38 to 48 felt a bit verbose and off the point.

- Line 49: "Most previous attempts... focused on developing countries..." This is not correct; many studies have focused on the US.

- Line 53-55: Consider the study, by Zhang et al.; doi: 10.5194/acp-16-9533-2016

- Line 68: remove "previous"

- Line 81: define AB32

[Figure]

- Line 82: Better explain "step function constraint"

- Line 88-89: How do projected energy consumption and population compare to current levels?

- Line 93: Change to "overview"

- Define acronyms upon first mention, e.g. EMFAC on line 105

- Line 117: Define VMT

- I found figure 2 hard to follow and not very informative. In the figure, I do not see where the 4 km2 resolution is achieved or the 2010 emissions inventory is used. I would recommend simplifying this figure into a version that better conveys the overall process, without showing every step included in the algorithm.

- Line 129: Is there a reference for the 2010 inventory?

- Line 136: Is 2050 meteorology being used? If so, what is the source of this data?

- Line 148: This appears to be a second scaling factor, beyond that just described in equations 1 and 2. This can be made clearer in the sentence.

- Lines 148-151: It seems that there are 2 projections being used, (1) the projection from EMFAC (which also accounts for policy) and (2) the CA-TIMES projection. Are both projections fully compatible?

- Line 280: change 2050 to 2010

- Line 318: Define ROGs

- Figure 10: This figure should use the same scale on all panels to adequately contrast the emissions projected under the BAU and GHG-Step scenarios. It would also be useful to map the difference between both scenarios, as well as the difference between each and the 2010 emissions inventory. In its current form, the figure is not very informative.

- The same observations mentioned above apply to figures 11-15.

- Lines 572-573: I'm not sure if this is clear; improving the figures would help.

- Figure 16: What are the units on the color scale?

- Lines 646-649: These sentences are unclear. Which are minor and major pollutants? this is not a typical classification.

- Lines 656-657: Cite literature supporting this.

- Lines 659-660: What is the rational for this statement? PM0.1 is a component of PM2.5, how are the health benefits amplified?

- Figure 17: The net total marker could be made clearer.

- Figure 18: The quality of this figure should be greatly improved. Label the panels, make the scale coloring uniform among them. What units are being used for NOx? This shows the change with respect to BAU or 2010?

- Line 706-707: This sentence is unclear. How is the effect immediate?

- Line 11: What is meant by "second or higher rounds of emissions controls"?

---

## Author Comment (AC1) · 26 Oct 2017

Response to Editor and Referee Comments on "Estimating Criteria Pollutant Emissions Using the California Regional Multisector Air Quality Emissions (CA-REMARQUE) Model v1.0" by Christina B. Zapata et al.

Response to Editor Comments

Comment 1. In order to comply with GMD policy, your manuscript must include the name and version number of the model in its title, the model code must be made avail-

able, and the code availability should be described in your "Code and/or Data Availability" section. If the original model code has not been modified for this manuscript, I think it would still be appropriate to mention the availability of the version of the model you used.

Response 1. We have added the model version number to the title as suggested.

Response to comments by Anonymous Referee #1

————- General comments: ————- In this manuscript, the authors use the CA-REMARQUE method to develop 4km gridded emission inventories for the year 2050. Two inventories are generated: a baseline inventory and one in which GHG emissions are reduced by 80% relative to 1990 levels. The scenarios were developed using the CA-TIMES model. The authors describe their approach for translating the CA-TIMES projections into criteria pollutant values based on a variety of sector-specific procedures. The authors then examine the resulting 4 km inventories and highlight differences in scenario-, sector-, and pollutant-specific emission trends. Furthermore, they compare the changes in emissions to those of a more generalized, expert-driven approach and suggest that considering California specific conditions (e.g., regulatory environment, existing stock, renewable resources) yield very different changes in emissions than a more generalized approach found in the literature.

Scientific significance: This work tackles an important objective - developing emissions inventories that can be used to examine the air quality implications of specific energy system scenarios. While others have also tackled this problem, this work adds important detail. However, because the literature review is very limited, the authors are not able to explicitly identify how they advance the science. (see "Specific Comments" for suggestions about expanding the literature review)

Scientific quality: I believe that the scientific approach and methods are underlying the work presented here are valid.

[Figure]

Scientific reproducibility: The authors do an very good job of describing the process that they used to develop future-year inventories from energy system modeling results. While replicating the work for this or another state or geographic region would undoubtedly be a large and difficult task, that difficulty would not be due to lack of information on the method.

Presentation quality: In general, I feel as if this manuscript is well written and that the experimental design and results reinforce the arguments presented by the authors. Nonetheless, I think the presentation quality could be improved substantially if the graphics provided in the results section were revised. One particular area of improvement is in the graphics that map PM emissions for both scenarios. I believe the intent of these graphics was to illustrate (i) the ability to develop spatially explicit inventories, and (ii) the changes in these emissions from one scenario to another. By using different scales on each image, however, the differences are not readily apparent. I suggest using the same scale or perhaps showing a graphic for the business as usual case and another with deltas associated with the GHG mitigation case. The stacked bar graphs that showed pollutant specific changes were also confusing and it was not clear what stacking of percents was intended to indicate. Those data could be much more easily presented and compared using a table.

———– Specific comments: ———– Comment 1: I feel that the manuscript has several deficiencies, and that it could be improved substantially if these deficiencies are addressed. Please see the "Specific Comments" below. My main concern is related to the literature review which could be expanded. I suggest that the authors include addition studies that examine the emissions or air quality impacts of alternative policy scenarios. A few for the U.S. are listed below (although I do not think that all of these necessarily need to be referenced): * Keshavarzmohammadian A, DK Henze, and JB Milford (2017). Emission impacts of electric vehicles in the US transportation sector following optimistic cost and efficiency projections. Envir. Sci. Technol., 51(12), 6665-6673. * Loughlin DH, WG Benjey, CG Nolte (2011). ESP v1.0: methodology for

exploring emission impacts of future scenarios in the United States. Geosci. Model Dev., 4, 287-297. * Ran L, DH Loughlin, D Yang, Z Adelman, BH Baek, and CG Nolte (2015). ESP v2.0: enhanced method for exploring emission impacts of future scenarios in the United States - addressing spatial allocation. Geosci. Model Dev., 8, 1775-1787. * Rudokas J, PJ Miller, MA Trail, and AG Armistead (2015). Regional air quality management aspects of climate change: Impact of climate mitigation options on regional air emissions. Environ. Sci. Technol., 49(8), 5170-5177. * Trail MA, AP Tsimpidi, P Liu, K Tsigaridis, Y Hu, JR Rudokas, PJ Miller, A Nenes, and AG Russell (2015), Impacts of potential CO2-reduction policies on air quality in the United States. 49(8), 5133-5141.

Similar to this manuscript, the Loughlin et al. 2011 paper also illustrates the development of region-, sectoral- and pollutant-specific implications of a climate policy and discusses how an energy system model's emission projections can be used to develop inputs to air quality modeling. Ran et al. (2015) build on that by adding a land use change compontent to spatially re-allocate emissions for some sectors. This manuscript would benefit greatly by a comparison to the Loughlin et al. and Ran et al papers. In such a comparison, I feel that the work presented here has much to add. For example, it explicitly tackles a state's regulations, goes into much greater detail regarding specific sectors (e.g., nonroad and marine), tackles the siting of new biomass-related emission sources, examines speciated emissions changes from fuel switching (e.g., fossil to biofuels), incorporates consideration of the PM benefits of regenerative braking, and seeks to examine the impact of controls on particle size distribution. Described in the context of these advances, I think the merits of the work presented here are much clearer and point to areas where other analyses could be improved.

Response 1: We agree that an expanded literature review describing the latest work and findings in field will strengthen the study and help put the results in context. We have provided a brief description of these studies and cited the references in the introduction section of the revised paper.

Comment 2: I have an additional concern that is perhaps less critical. My concern is related to the presentation of the scenarios themselves. The first scenario is relatively easy to understand as it is just a baseline or business as usual scenario. Nonetheless, California has a very unique energy system, so the underlying trends and dynamics there may not be readily apparent to readers. The second scenario is a GHG reduction scenario, although how the GHG constraint is implemented is not completely clear. Since the authors only compare emissions between a base year, 2010, and a future year, 2050, they have chosen to show the CA-TIMES results only for those years. To convey the scenarios and underlying dynamics more fully, however, I believe the authors should show several model outputs for the period from 2010 *through* 2050, not just in 2010 and 2050. These outputs would include at a minimum, and for both scenarios: (i) CO2 emissions by sector, and (ii) electricity production by technology, and (iii) energy system fuel use. Additionally, displaying fuel used or technologies adopted in the transportation and transportation sub-sectors could be of interest but are not necessary. Stacked area plots would be one form of presenting these graphics. The sectoral GHG graphic would be particularly useful in understanding how the GHG policy is represented in the model. For example, is it a constraint represented only from 2050 onward as the text seems to imply? If so, how did the model respond to such a shock, and did it begin to make structural changes to the energy system with foresight or respond in more of a myopic way? These results wouldn't necessarily need to be in the main body of the manuscript, and instead could be in the supplemental information.

Response 2: The three CA-TIMES predictions for CO2 emissions, electricity production and energy/fuel for each scenario are described in detail by Yang, Yeh et al. (2015), mainly Fig. 3, Fig. 8, and Fig. 10 or in the free version of the entire report (Yang, Yeh et al. 2014). An explicit reference to these publications is now provided on line 108 of the revised manuscript, and a summary figure for the intermediate year 2030 is now included in the SI. Fuel consumption/production was extracted from CA-TIMES results for 2050 and provided in the SI to assist in explaining the emission results.

The GHG-STEP is a step change in the emissions requirement, but because the model has perfect foresight and is optimizing (minimizing) the energy system cost over the entire modeling period from 2010 to 2050 (with a 4% discounting factor), it makes investment decisions that are necessary to ensure that the 2050 targets are met. Because technologies have finite lifetimes and cannot take over their respective markets instantaneously (in 2050), investments in low-GHG technologies start slowly and grow to reach the required market share needed to meet the targets. These points have been clarified on line 102 of the revised text.

Additional substantive comments are listed below, preceded by page number:

Comment 3: 55: The text refers to other modeling efforts as over-simplifying the California economy. This statement, I think, implies that CA-TIMES includes a much more detailed representation of the economy. I advise care with that description, however. I suspect that CA-TIMES respresents the "energy economy" only, which is only a portion of the larger economy, and may represent interactions with the rest of the economy via simple elasticities for energy demands or perhaps with a function that links to regional GDP. Many readers will assume that a model of the economy would take the form of a Computable General Equilibrium model, which would represent things like tradeoffs between labor and capital, employment, household income, etc.

Response 3: We thank the reviewer for pointing out this over-generalization of our description for CA-TIMES, which models California's energy system, not the rest of the economy. We have modified the text on line 55 of the manuscript to clarify that CA-TIMES is not a CGE model and we specify demands for energy services exogenously, which are not affected by technology choices.

Comment 4: 76: I suggest adding additional information to the description of CA-TIMES. For example, listing the sectors included would be helpful, as would listing the pollutants included in the model. Does it just represent $CO_2$, or does it add other GHGs? What about air pollutants? Are demands fixed or elastic? How the model

handles the rest of the country and the rest of the world? How trade with region(s) outside of CA is considered and constrained? Answers to some of these questions are alluded to later in the paper (as we see what CA-TIMES has produced and how it is used), but it would be very useful to have these types of questions answered explicitly when the model is introduced.

Response 4: The CA-TIMES description has been enhanced starting on line 86 of the revised manuscript. Supply sectors are electricity and resources and fuel supply. Demand sectors are transportation, residential, commercial, industrial and agricultural. CA-TIMES includes emissions of $CO_2$, methane and $N_2O$, but not air pollutants (these are added in the current manuscript). Demands for energy services are fixed. Imports are allowed from out of state (e.g. oil and natural gas) with specified costs that are considered in the overall optimization scheme. Renewables and Biomass within the state and outside the state are handled separately with import levels once again chosen to optimize total costs.

Comment 5: Fig 8: What is driving the reduction in commercial energy usage in 2050 GHG-Step? Is it elastic demands? More efficient technologies? Or is there commercial (on-site) solar power that isn't being shown?

Response 5: The text on line 427 of the revised manuscript has been modified to state that commercial energy reduction is due to efficiency and electrification of end uses.

Comment 6: Fig 10: At first glance these look the same. It isn't until you examine the legend that it is clear that there has been a 2/3 reduction. I suggest having the scale the same on all the graphs. Alternatively, you could show the 2050 emissions and another graph showing deltas (e.g., where they increase or decrease). The latter approach may be more useful in conveying air pollutant emission co-benefits (or dis-benefits).

Response 6: Figure 10 has been revised to show the BAU and the delta emissions as suggested by the reviewer.

Comment 7: 689 - Stacking percentages are difficult to interpret? Was a single % change then apportioned to components? If not, it may not be appropriate to stack these values. Perhaps you could show the actual quantities? I feel like a table would convey the information better.

Response 7: The bar chart in Figure 17 was designed to indicate how each emission sector or source category is contributing to a percentage change in the nominal emissions. The text on line 700 of the revised manuscript has been added to better explain the figure.

Contributions below 0% indicate emissions reductions, while contributions above 0% indicate emissions increases. Each of these changes represents the statewide average for the sources within the indicated sector. Note that the changes within each sector may not be uniform across the entire state. The net change in total emissions is indicated by the black horizontal line for each species.

A table of the results summarized in Figure 17 has been added to SI.

———– Editorial comments and corrections: ———– My other comments, which are more editorial in nature, are listed below. I hope that these are helpful in revising what I feel is a very interesting manuscript. Line #: comment

Comment 8: 54: Editorial suggestion - Here and throughout, avoid using "/". Also, where you have "and/or" I suggest just using "or", which is not necessarily mutually exclusive.

Response 8: Text throughout the manuscript changed to use "or" instead of "and/or".

Comment 9: 86: CAFE should be spelled without the accent mark on the E

Response 9: Corrected.

Comment 10: 87: References to the regulations affecting CA would be helpful

Response 10: References added.

Comment 11: 105: The statement is made that "vehicular emissions for the year 2050 were extrapoled", which left me wondering how. I see that you explain this later in the paper, but is there a way to reorganize so the reader doesn't ask this question, or, if they do, the answer if provided sooner? I had a similar concern with the text "vehicular activity and fuel consumption splits were applied..." in line 108.

Response 11: The text in Section 2.1 of the revised manuscript has been revised to more clearly explain how emissions were extrapolated and fuels were split when this issue is first introduced.

Comment 12: Figure 2: The abbreviation EIC is used prior to being defined

Response 12: Emissions Inventory Code (EIC) now defined in Figure 2.

Comment 13: Figure 2: Minor suggestion: for the "2050 CA-TIMES Scenario on-road Emissions" box, I suggest renaming this "On-road emissions consistent with a 2050 CA-TIMES Scenario".

Response 13: Figure 2 has been extensively revised in response to another reviewer comment. Text now reads "CA-TIMES 2050 Projections for vehicle miles traveled (VMT), fuel, and fleet composition statewide.

Comment 14: 128: A meteorological scaling factor is mentioned, but it isn't clear what is assumed about meteorology in the future.

Response 14: Text on line 168 revised to clarify that temperature and relative humidity are downscaled to 4km resolution in 2054.

Comment 15: 128: It appears that the methodology does not explicitly consider expansion of existing roadways or the addition of new roadways. Similarly, I did not see a mention of the impact of land use change. If not, perhaps they could be mentioned in the discussion section as ways in which the analysis could be expanded?

Response 15: The current analysis assumes that no new major highways will be built

in California and population growth is accommodated partially through increased urban density such that traffic volumes increase uniformly across the transportation network. We recognize these assumptions are simplistic but a previous study of smartgrowth in the San Joaquin Valley indicated that more detailed accounting of population growth had minimal impact on air quality. These points are clarified in the updated discussion section of the paper as suggested by the Reviewer.

Comment 16: 145: There are two equations listed as (1). I suggest making each its own.

Response 16: Corrected.

Comment 17: 155: Please provide definitions for the vehicle sub-categories

Response 17: Clarified on line 191 of the revised manuscript, with an explicit reference to Table S2.

Comment 18: 209: Replace "_" with "approximately"?

Response 18: Corrected.

Comment 19: 223: "Fueled" misspelled?

Response 19: Corrected.

Comment 20: 345: Perhaps a better way to state "CA-TIMES finds that it is too expensive to adopt biomass-based fuels for ships in the GHG-Step scenario in 2050" is that the model identifies other approaches for meeting the GHG target more cost-effectively?

Response 20: Revised as suggested.

Comment 21: 376: awkward

Response 21: Reworded.

Comment 22: 388: Were wood and distillate held constant, or were they allowed to

compete in CATIMES... but just not allowed to increase?

Response 22: Clarified on line 426. These sources were allowed to compete in CA-TIMES subject to the constraint that they could not increase above 2010 levels in order to maintain compliance with current air quality regulations.

Comment 23: Fig 9: Too many significant figures are shown in the table. Perhaps limit to 3 for each value shown? Response 23: Significant figures reduced to 3 as suggested.

Comment 24: 431: CA-GREET1.8b is argued to have the highest accuracy. But it isn't clear what that means or to what it is being compared to evaluate accuracy.

Response 24: Text on line 480 revised to clarify

An inter-comparison study between GREET1.8, GREET 2014, and CA-GREET2.0 showed that theCA-GREET1.8b model had the best agreement with emissions rates from approximately 30 biomass plants operating on wood residue in California. (California Air Resources Board 2011, US Environmental Protection Agency 2014).

Comment 25: 447: Replace "predicted" with "determined"? CA-TIMES and similar models aren't typically referred to as predictive models. Instead, they are used to evaluate how technology and fuel choices play out under particular scenarios and assumptions.

Response 25: Revised as suggested.

Comment 26: 450: Do you know the amount of reduced usage vs. switching to imports?

Response 26: Based on purely the crude oil supply numbers shown in Figure S6, crude oil supply reduces by 44% (2,400PJ down from 4300PJ in 2010) in the GHG-Step scenario. We're not switching to imports as much as eliminating crude extraction within California (0PJ in 2050 GHG-Step versus 1500 in 2010). This is likely due to

carbon constraint and costs of extraction determined in CA-TIMES relative to importing. Imports would also drop slightly by 14% (down to 2400PJ relative to 2800PJ in 2010) for this scenario. Additional information added on line 497.

Comment 27: 453: Where "Assumed to increase" is used, I think the authors are referring to a model result, not an assumption. Perhaps reword to clarify.

Response 27: Corrected.

Comment 28: 711: The legend for Fig 18a should include at least one more significant digit to be more comparable with the other graphics.

Response 28: Fig. 18 revised.

Comment 29: 744: You should probably be clear that the CA-TIMES model "code" isn't part of this package (assuming that it is not).

Response 29: Clarified.

Comment 30: 734: "does not have significant impact on criteria..." I understand the point, but this is worded awkwardly.

Response 30: Revised.  

Response to comments by Anonymous Referee #2

The authors present an ambitious modeling effort to project air pollutant emissions from multiple sectors under differing climate policies at high resolution across California. In doing so, this study carries out a significant effort to simulate changes to technology, fuels and human activity for varying sources based on multiple models, datasets and careful assumptions. The paper describes a methodology that goes beyond prior attempts to project pollutant emissions changes associated with GHG mitigation strategies. The methodology and model developed will be a valuable contribution to the air quality and climate modeling communities. Additionally, the manuscript is well written. I believe the paper should be considered for publication in GMD. However, I have some

concerns with the manuscript in its current form. These issues must be addressed before I can recommend publication. My comments are described below:

Comment 1: - My largest concern relates to the presentation of results under section 3 Results and Discussion. Although the authors describe the model's results and discuss some interesting findings, the section (particularly the plots and visualizations) should be improved to better communicate the study's results. Figures 10-15 are not informative. Figures 16 and 18 are unclear. The quality of all figures with maps in the section could be improved. Some specific comments are provided below. I encourage the authors to improve the manuscript's overall discussion of results.

Response 1: Figures 10-15 were revised to show the spatial detail and location of various emission sources as well as indicate changes between scenarios. Figures 16 and 18 captions and scaling are clarified. The legend scaling for all these figures have been updated to improve clarity and ease of scenario comparison.

Comment 2: -Although the study focuses on California, and the depth with which the authors model projected emissions within the state is a strength, the manuscript would benefit from including a discussion of the potential benefits and challenges associated with applying the emissions modeling methods used to other regions beyond California or at a national scale. Given the specificity of the analysis, it is difficult to identify which elements may be extended to other locations or to Policy/GHG/Energy projections developed with other models beyond CA-TIMES.

Response 2: The basic calculations demonstrated in the current manuscript to estimate criteria pollutant emissions that are consistent with energy policies can be extended outside California, but the tools and data sources needed for the analysis would need to be modified as appropriate for each new region. Within the United States, the mobile sector analysis would need to use the MOVES model created by US EPA rather than the EMFAC model created by the California Air Resources Board (CARB). The spatially-resolved based inventory would be produced from the National Emissions In-
ventory (NEI) rather than the inventory maintained by CARB.

The input data needed for other sectors may be more difficult to obtain outside of California. Changes to off-road emissions would need to be estimated following procedures similar to those employed in the CARB off-road VISION model. Effort would be needed to estimate how changes to marine fuel sources would influence emissions at major ports. Studies would need to be conducted describing potential locations for new facilities producing low-carbon fuels and the resulting emissions from those facilities. All of these studies have could be carried out for regions outside of California given sufficient time and resources.

Comment 3: - Line 17: Change "play" to "plays"

Response 3: Revised.

Comment 4: - Cite references to strengthen the statements in lines 38-40, 40-41 and 42-44.

Response 4: Section removed since verbose (Comment 5).

Comment 5: - The paragraphs from line 38 to 48 felt a bit verbose and off the point.

Response 5: Removed.

Comment 6: - Line 49: "Most previous attempts: : : focused on developing countries: : :" This is not correct; many studies have focused on the US.

Response 6: Revised to discuss US studies more thoroughly.

Comment 7: - Line 53-55: Consider the study, by Zhang et al.; doi: 10.5194/acp-16-9533-2016

Response 7: Added.

Comment 8: - Line 68: remove "previous"

Response 8: Removed.

Comment 9: - Line 81: define AB32

Response 9: Defined AB32 as California Assembly Bill 32, the Global Warming Solutions Act on line 97.

Comment 10: - Line 82: Better explain "step function constraint"

Response 10: Text revised to explain step function more clearly on line 99.

In the GHG-Step scenario a "step" GHG emissions function constraint is applied in which a constant 2020 cap is held until 2050, and then an 80% reduction is applied from 2050 onward. This allows the model to adopt strategies that lower GHG emissions prior to 2049 if those strategies minimize costs. This 2050 GHG constraint does not shock the energy system because the CA-TIMES model has perfect foresight and optimally minimizes the energy system cost (with a 4% discount factor) over the entire period from 2010 to 2050 making investment decisions to meet targets. Also, CA-TIMES investments in low-GHG technologies start slowly and grow to reach the required market share to meet the targets since technologies have finite lifetimes and cannot take over respective markets instantaneously.

Comment 11: - Line 88-89: How do projected energy consumption and population compare to current levels?

Response 11: Text on line 115 revised to read

CA-TIMES predicts total annual energy consumption in California for the year 2050 to be 8,763 PJ in the BAU scenario and 7,679 PJ in the GHG-Step scenario (reference value for 2010 is approximately 7,500 PJ).

Text on line 647 revised to read

Population growth from approximately 37.4 million in 2010 to approximately 50.4 million in 2050 was assumed to be identical under the BAU and GHG-Step scenarios yielding virtually identical spatial distributions for both scenarios.

Comment 12: - Line 93: Change to "overview"

Response 12: Fixed.

Comment 13: - Define acronyms upon first mention, e.g. EMFAC on line 105

Response 13: EMissions FACtor now defined on line 133 of the revised manuscript.

Comment 14: - Line 117: Define VMT

Response 14: Vehicle Miles Traveled (VMT) defined on line 138 of the revised manuscript.

Comment 15: - I found figure 2 hard to follow and not very informative. In the figure, I do not see where the 4 km2 resolution is achieved or the 2010 emissions inventory is used. I would recommend simplifying this figure into a version that better conveys the overall process, without showing every step included in the algorithm.

Response 15: As requested, Figure 2 has been revised to provide an overview of the mobile source process that should be easier to follow.

Comment 16: - Line 129: Is there a reference for the 2010 inventory?

Response 16: No published reference is available but the inventory can be provided upon request by the staff at the California Air Resources Board.

Comment 17: - Line 136: Is 2050 meteorology being used? If so, what is the source of this data?

Response 17: Yes, 2050 meteorology is being used as described on line 167 of the revised manuscript. Comment 18: - Line 148: This appears to be a second scaling factor, beyond that just described in equations 1 and 2. This can be made clearer in the sentence.

Response 18: Clarified.

Comment 19: - Lines 148-151: It seems that there are 2 projections being used, (1)

the projection from EMFAC (which also accounts for policy) and (2) the CA-TIMES projection. Are both projections fully compatible?

Response 19: EMFAC accounts for population growth and emissions changes that are required by existing air quality rules and regulations through 2050. CA-TIMES accounts for additional changes that will be required to comply with state GHG targets but which have not yet been placed into emissions rules and regulations. The two projections are compatible since we have not double-counted any emissions reductions. This has been clarified on line 186 of the revised text.

Comment 20: - Line 280: change 2050 to 2010

Response 20: Thank you. Corrected.

Comment 21: - Line 318: Define ROGs

Response 21: Reactive Organic Gases (ROGs) defined on line 169.

Comment 22: - Figure 10: This figure should use the same scale on all panels to adequately contrast the emissions projected under the BAU and GHG-Step scenarios. It would also be useful to map the difference between both scenarios, as well as the difference between each and the 2010 emissions inventory. In its current form, the figure is not very informative.

Response 22: Figure revised as suggested.

Comment 23: - The same observations mentioned above apply to figures 11-15.

Response 23: Figures revised as suggested.

Comment 24: - Lines 572-573: I'm not sure if this is clear; improving the figures would help.

Response 24: Revised figures illustrate the trend more clearly.

Comment 25: - Figure 16: What are the units on the color scale?

Response 25: They are PM2.5 mass emissions in $\mu$g m-2 min -1 of production at each facility. Clarified.

Comment 26: - Lines 646-649: These sentences are unclear. Which are minor and major pollutants? this is not a typical classification.

Response 26: Text clarified on lines 703-708.

Comment 27: - Lines 656-657: Cite literature supporting this.

Response 27: Added.

Comment 28: - Lines 659-660: What is the rational for this statement? PM0.1 is a component of PM2.5, how are the health benefits amplified?

Response 28: The text on line 716 has been clarified to read

Recent epidemiology results indicate that PM0.1 is associated with mortality in the California Teachers Study (Ostro, Hu et al. 2015). Enhanced PM0.1 emissions reductions could amplify the potential health benefits of the future GHG-Step scenario beyond the level expected from reductions in PM2.5 emissions reductions.

Comment 29: - Figure 17: The net total marker could be made clearer.

Response 29: Done.

Comment 30: - Figure 18: The quality of this figure should be greatly improved. Label the panels, make the scale coloring uniform among them. What units are being used for NOx? This shows the change with respect to BAU or 2010?

Response 30: Figure modified as requested.

Comment 31: - Line 706-707: This sentence is unclear. How is the effect immediate?

Response 31: Revised text to indicate the changing emissions patterns will have a direct effect on population exposure.

Comment 32: - Line 11: What is meant by "second or higher rounds of emissions controls"?

Response 32: Text at line 768 revised to state more clearly Similar efforts will be required to analyze the effects of GHG mitigation strategies on criteria pollutants in other highly-populated regions that have already moved beyond simple emissions regulations banning obvious sources of air pollution.

References California Air Resources Board. (2011, May 28, 2015). "Facility Search Engine Tool." Retrieved 6/5/2015, 2015, from www.arb.ca.gov/app/emsinv/facinfo/facinfo.php. Ostro, B., J. Hu, D. Goldberg, P. Reynolds, A. Hertz, L. Bernstein and M. J. Kleeman (2015). "Associations of mortality with long-term exposures to fine and ultrafine particles, species and sources: results from the California Teachers Study Cohort." Environ Health Perspect 123(6): 549-556. US Environmental Protection Agency. (2014, 02/24/2014). "eGRID. Nineth edition with year 2010 data (Version 1.0)." 9th. Retrieved 6/5/2015, 2015, from www.epa.gov/cleanenergy/energy-resources/egrid/. Yang, C., S. Yeh, K. Ramea, S. Zakerinia, D. McCollum, D. Bunch and J. Ogden (2014). Modeling Optimal Transition Pathways to a Low Carbon Economy in California: California TIMES (CA-TIMES) Model. Davis, CA., Institute of Transportation Studies, University of California, Davis. Yang, C., S. Yeh, S. Zakerinia, K. Ramea and D. McCollum (2015). "Achieving California's 80% greenhouse gas reduction target in 2050: Technology, policy and scenario analysis using CA-TIMES energy economic systems model." Energy Policy 77: 118-130.

---

## Author Response (AR2)

Reviewer 1

I appreciate the authors' efforts to respond to my comments and feel that my concerns have been adequately addressed in the revised manuscript. I have a few additional suggestions below that may further improve the manuscript before final publication.

Comment 1- Figures 10-15. While greatly improved, I would still recommend using whole numbers (and perhaps a smaller number of levels) on the color scales to improve the visuals.

Response 1 – The number of color levels has been reduced from 14 to 10 in Figures 10-15, and the max/min values have been adjusted to values that generate more logical boundaries for each concentration level.

Comment 2 - I would recommend adding the explanation presented in "Response 2" of the authors' response, related to methods applicability outside of California, to the manuscript text.

Response 2 – This text already included starting at line 766 of the revised manuscript.

Comment 3 - For some reason, the citations in the updated text have 2 authors listed.

Response 3 – corrected.

Comment 4 - Response 28: While Ostro et al. show an association between ischemic heart disease and ultrafine PM, they do not argue that the association is necessarily stronger than that with PM2.5. The evidence of a higher toxicity associated with PM0.1 is still limited, and I believe the conclusion that "PM0.1 emissions reductions could amplify the potential health benefits … beyond the level expected from PM2.5 emissions reductions" may not be very strongly substantiated.

Response 4 – a number of toxicology studies have demonstrated that ultrafine particles have greater toxicity per unit mass than fine particles [1-4]. These studies have been cited in the revised manuscript to support the statement that PM0.1 emissions reductions are potentially important.

References:

1. Donaldson, K., et al., *The pulmonary toxicology of ultrafine particles.* Journal of Aerosol Medicine-Deposition Clearance and Effects in the Lung, 2002. **15**(2): p. 213-220.
2. Donaldson, K., et al., *Ultrafine particles.* Occupational and Environmental Medicine, 2001. **58**(3): p. 211-+.
3. Elder, A., et al., *Translocation of inhaled ultrafine manganese oxide particles to the central nervous system.* Environmental Health Perspectives, 2006. **114**(8): p. 1172-1178.

4.     Kreyling, W.G., M. Semmler, and W. Moller, *Dosimetry and toxicology of ultrafine particles.* Journal of Aerosol Medicine-Deposition Clearance and Effects in the Lung, 2004. **17**(2): p. 140-152.

Reviewer 2

The authors made great strides to address my concerns from my review of their original submission to GMDD. My remaining suggestions are largely editorial. I have only one content-related suggestion:

Content suggestion:

Comment 1 - Page 3 / line 89:  I still feel that the process of interpreting the results by the reader would be improved greatly if two graphics were added to the text of the manuscript... stacked bar or area charts of sectoral GHG emissions for the BAU and GHG-Step scenarios. These graphics would allow the reader to understand how the model is responding to the constraint, which is important since those responses are what drive the emissions changes highlighted later in the manuscript. While the revised text has been modified to refer to where to find such information, and while the supplemental info now includes the GHG-Step plot, I think both these graphics should be explicitly included in the body of the manuscript. I don't think this is a required change, per se, but it would improve the manuscript.

Response 1 – graphic added as requested.

Comment 2 - As an aside, the authors suggest that the GHG-Step constraint being implemented at a single future year (as opposed to phased in) does not result in a discontinuity in the model's response. When I look at the GHG-Step graphic, I do see a discontinuity in 2050, so I think the authors' revised text indicating that there is no discontinuity is not correct or is at least too strong.

Response 2 - The CA-TIMES model has foresight and changes are occurring in the years 2040-49 to prepare the base needed for 2050.  Actions in this time frame include decarbonizing the grid, the important task of stock turnover in vehicles and stock turnover in building appliances.  100% of sales of fully clean and electrified vehicles and appliances in 2050 would not be enough to decarbonize these sectors if changes were not already occurring between 2040-49.  100% of sales in a given year doesn't translate to 100% of the fleet being converted to that technology.  Vehicles have a 15 year lifetime in CA-TIMES, therefore 1/3 of the fleet changes every five years.  Similar trends are true for appliances.

We have modified the description in the manuscript to read

"This 2050 GHG constraint causes aggressive change over the period 2040-49 but does not shock to the energy system in 2050 because the CA-TIMES model has perfect foresight and optimally minimizes the energy system cost (with a 4% discount factor) over the entire period from 2010 to 2050 making investment decisions to meet targets."

Editorial suggestions:

I feel that the authors' language and choice of words could be tightened. For example, below are some particular items:

Comment 3 - Page 1/line 9 "sophisticated programs" - It is unclear what is meant by "programs." Computer programs? Management programs? Whether they are "sophisticated" also seems very subjective. Perhaps "comprehensive" would be a better adjective (although I recognize that this can also be subjective).

Response 3 – "Sophisticated programs" changed to "sophisticated emissions control programs". The authors assert that California has some of the most sophisticated GHG emissions control programs in the world, and hence the statement is not subjective.

Comment 4 - Page 1/line 11 "aggressive GHG reduction" - One person's "aggressive" is another person's "moderate," and another person's "lenient." I think these terms are best used when comparing policies (e.g., policy A is more lenient than policy B). I suggest just referring to this as a GHG policy. If you feel it is aggressive, perhaps you could cite a reference to this interpretation?

Response 4 – "…aggressive GHG reduction…" replaced with "…80% GHG reduction…".

Comment 5 - Page 1/line 13 "necessarily include changes in..." - I think "necessarily" is too strong here. Also, I would add "across economic sectors" at the end.

Response 5 – changed as requested.

Comment 6 - Page 1/line 23 "manifests most notably through a comparison" - I don't think this use of manifest is correct... This reads as if the act of comparing the results leads to the behavior occurring, which I do not believe is the intent. Perhaps "apparent most notably through a comparison"...

Response 6 – changed as requested.

Comment 7- Page 1/line 31 "debating optimal strategies" - "optimal" typically refers to the mathematically least cost option. Policymakers don't generally think in those terms. I suggest "debating alternative strategies", "debating candidate strategies", or "debating cost-effective candidate strategies".

Response 7 – changed to "debating cost-effective candidate strategies" as requested.

Additional editorial suggestions:

Comment 8 - Page 29/ line 689 - "each ... experience" should be "each ... experiences"

Response 8 – changed as requested.

Comment 9 - Page 33 / line 773 - This refers to having evaluated multiple GHG policies, but you really only evaluated one here.

Response 9 – We evaluated a single scenario that includes multiple policies. No changes made in response to this comment.

Comment 10 - Page 1/line 21 "with obvious implications" - At least one style guide that I have read suggests avoiding words such as "obvious" and "clearly".

Response 10 – deleted "obvious".

Comment 11 - Page 1/line 25- Page 1/line 26 "PM2.5", "PM0.1" should use subscripts

Response 11 – changed as requested throughout the manuscript.

Comment 12 - Page 1/line 25 "vs." Spell out "versus"?

Response 12 – changed as requested throughout the manuscript.

Comment 13 - Page 1/line 32 "including (among other things)..." - This is redundant since "including" implies you may not be listing all those things.

Response 13 – deleted "(among other things)".

Comment 14 - 15/360 - Not necessary to use apostrophe on "CA-TIMES"

Response 14 – deleted apostrophe as requested.

Throughout:

Comment 15 - I don't recall ever seeing in-text citations of this format... two authors, followed by "et al" (e.g., Curly, Moe, et al., 2015). Please check the journal requirements. Usually these in-text citations would just be: (e.g., Curly et al., 2015).

Response 15 – shortened to one author in citations.

Comment 16 - Commas are not used in a consistent manner in the abstract and introduction sections

Response 16 – corrected.

Graphics:

Comment 17 - For Figure 9: I think this is OK, but it is not necessary to include the data under the table. The data could be provided in the Supplemental Information instead.

Response 17 – data table removed from the bottom of Figure 9 as requested.

Comment 18 - For Figures 10 through 15: I greatly prefer the formatting of these figures to the versions in the original submission. I have only a minor suggestion. It would be helpful to readers to more explicitly identify the right hand column as depicting deltas. Also, having the units on the graphic itself would be useful. Finally, I think it should be more clear that the column on the right is comparing both results for the year 2050. One possible suggestion would include the following changes:

(i) change the heading over the left column to:

2050 BAU (ug m-2 min-1)

(ii) and the right column to:

2050 GHG-Step minus 2050 BAU (ug m-2 min-2)

It this is too much text for the right-hand column, perhaps you could put the units under the column name?

Response 18 – changed as requested.

---

## Author Response (AR3)

Editor Comments

Comment 1- I would like to point the authors to the program code and data access policy of the GMD journal as outlined in https://www.geoscientific-modeldevelopment.net/about/manuscript_types.html. Main objective of the policy is to guarantee reproducibility of the model result presented in paper. In accordance of this objective I would like to encourage to consider uploading your program code and data as supplementary material to ensure persistent access to the data. It would be also useful and interesting for the user to learn about the program environment the model code is running in and under what license the software can be used .

Response 1 –As requested, we have updated the Code and Data Availability Section to read

CA-REMARQUE was developed and executed in the Linux programming environment using standard shell scripts and FORTRAN programs compiled using the Portland Group software.  All of the data necessary to calculate changes to emissions inventories are published in full in the main text and supporting information section of the manuscript.  The output emissions datasets are available free of charge at faculty.engineering.ucdavis.edu/kleeman/.  The program code is currently being updated to use the latest version of the California EMFAC software and will be posted at faculty.engineering.ucdavis.edu/kleeman/ when complete.  Note that the CA-REMARQUE v1.0 model is separate from the CA-TIMES energy-economic model and the California EMFAC model.